# GAN You See Me? Enhanced Data Reconstruction Attacks against Split Inference

Ziang Li[1], Mengda Yang[1], Yaxin Liu[1], Juan Wang[1] [*], Hongxin Hu[2], Wenzhe Yi[1] and Xiaoyang Xu[1]

[1]Key Laboratory of Aerospace Information Security and Trusted Computing, Ministry of Education,
School of Cyber Science and Engineering, Wuhan University
[2]Department of Computer Science and Engineering, University at Buffalo, SUNY

## Abstract

Split Inference (SI) is an emerging deep learning paradigm that addresses computational constraints on edge devices and preserves data privacy through collaborative edge-cloud approaches. However, SI is vulnerable to Data Reconstruction Attacks (DRA), which aim to reconstruct users' private prediction instances. Existing attack methods suffer from various limitations. Optimization-based DRAs do not leverage public data effectively, while Learning-based DRAs depend heavily on auxiliary data quantity and distribution similarity. Consequently, these approaches yield unsatisfactory attack results and are sensitive to defense mechanisms. To overcome these challenges, we propose a **G**AN-based **LA**tent **S**pace **S**earch attack (*GLASS*) that harnesses abundant prior knowledge from public data using advanced StyleGAN technologies. Additionally, we introduce *GLASS++* to enhance reconstruction stability. Our approach represents the first GAN-based DRA against SI, and extensive evaluation across different split points and adversary setups demonstrates its state-of-the-art performance. Moreover, we thoroughly examine seven defense mechanisms, highlighting our method's capability to reveal private information even in the presence of these defenses.

## 1 Introduction

The emergence of Deep Learning (DL) has brought about a transformative impact on machine learning applications, granting them remarkable capabilities. To cater to the increasing demand for DL models on edge-side devices, various challenges related to performance arise. The growing size of model parameters poses a burden on resource-constrained edge devices. As a result, the concept of Machine Learning as a Service (MLaaS) has gained popularity as a solution. However, deploying DL services in the cloud, where APIs are provided to users and raw data is collected for service provisioning, raises concerns about potential data leakage. In this context, Split Inference (SI) has emerged as a promising alternative [Eshratifar et al., 2019; Banitalebi-Dehkordi et al., 2021; Kang et al., 2017; Matsubara et al., 2022; Hauswald et al., 2014]. SI involves splitting and deploying DNN models between the edge and the cloud, allowing the cloud's extensive computing and storage resources to be leveraged while ensuring that users only need to upload intermediate feature representations to protect the confidentiality of their original data.

However, recent studies have demonstrated that even with the use of intermediate feature representations, a malicious cloud server can still launch privacy attacks. Of particular concern is the Data Reconstruction Attack (DRA) [He et al., 2019; Singh et al., 2021; Yang et al., 2022], which represents the most severe violation of user privacy as it aims to reconstruct the user's inference

---

[*]Corresponding author: jwang@whu.edu.cn

37th Conference on Neural Information Processing Systems (NeurIPS 2023).

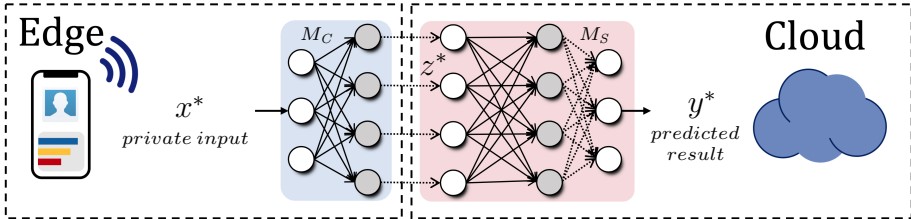

Figure 1: Split Inference.

instances. Existing DRAs suffer from critical flaws that significantly diminish their effectiveness. Optimization-based DRAs [He et al., 2019; Singh et al., 2021], for example, fail to effectively leverage public data, while Learning-based DRAs [He et al., 2019] heavily depend on the quantity of auxiliary data and require a high degree of distribution similarity between the auxiliary data and the inference data.

To address these limitations and enhance the impact of the attack, we propose the **G**AN-based **LA**tent **S**pace **S**earch attack (GLASS). This attack leverages the power of StyleGAN [Karras et al., 2019, 2020, 2021; Sauer et al., 2022] and fully capitalizes on the valuable prior knowledge embedded in public data. Additionally, we introduce *GLASS++*, an improved version that enhances attack stability and effectiveness.

We systematically evaluate the reconstruction performance of Optimization-based *GLASS* and Learning-based *GLASS++* on face data at different split points. Additionally, we thoroughly examine and analyze seven advanced defense mechanisms against DRA in SI. These mechanisms are categorized into three types: Clipping (Dropout Defense[He et al., 2020], DISCO[Singh et al., 2021]), Noise Addition (Noise Mask[Titcombe et al., 2021], Shredder[Mireshghallah et al., 2020]) and Feature Obfuscating (Adversarial Learning[Li et al., 2021], NoPeek[Vepakomma et al., 2020] and Siamese Defense[Osia et al., 2020]). Furthermore, we go beyond the traditional assumptions about adversary capabilities and extend our attack to heterogeneous data. Through our extensive experimentation, we achieve superior attack results across various split points and different adversary setups, successfully bypassing the employed defense mechanisms and compromising their effectiveness.

The key contributions of this paper are:

- We propose *GLASS* and *GLASS++*, which are enhanced DRAs combined with pre-trained StyleGAN models. This is the first instance of utilizing the latent space search characteristic of StyleGAN to develop DRAs specifically for SI. Additionally, we expand the practicality of our methods by designing attack strategies for various adversary settings.

- Through the utilization of advanced StyleGAN technologies, we exploit the rich prior knowledge present in public data, resulting in state-of-the-art reconstruction performance across different split points. Our methods outperform existing baseline attacks on multiple evaluation metrics, showcasing their superiority.

- We conduct a systematic evaluation and comparison of various DRAs against seven defense mechanisms. The results demonstrate that our methods effectively reveal sensitive information and undermine the robustness of the defenses.

## 2 Background and Related Work

### 2.1 Split Inference

Split Inference (SI) and Split Learning (SL)[Gupta and Raskar, 2018; Thapa et al., 2022; Poirot et al., 2019] have emerged as promising alternatives, which split and deploy DNN models on both the edge-side and the cloud side. In SI, a well-trained model $M$ is split into client model $M_C$ and server model $M_S$. An inference data $x^*$ is fed to $M_C$ to get an intermediate feature representation $z^* = M_C(x^*)$. $z^*$ is then transmitted to the cloud to execute $y^* = M_S(z^*)$. Finally, $y^*$ is returned to the edge-side to complete the inference process, as shown in Figure 1. Collaborative computing across edge-cloud devices facilitates the reduction of computing payload on the edge side. By transmitting

only the smashed data to the cloud side, a certain level of privacy protection can be ensured. Utilizing distributed inference/training protocols, SI and SL achieve an improved trade-off between utility and privacy.

## 2.2 Data Reconstruction Attacks on Split Inference

Data Reconstruction Attack (DRA) is one of the most powerful privacy attacks that focuses on reconstructing private inference data. The existing DRAs can be broadly categorized as Optimization-based and Learning-based. [He et al., 2019] introduced *regularized Maximum Likelihood Estimation* (rMLE), firstly treating DRA as an optimization problem. For an intermediate feature representation $z^* = M_C(x^*)$, they find the optimal sample $x$ which minimizes the posterior information from feature-level observation by reducing the Euclidean Distance between $M_C(x)$ and $M_C(x^*)$. Additionally, they adapt the Total Variation (TV)[Rudin et al., 1992] to represent the prior information derived from the distribution of natural images. Inspired by the deep image prior[Ulyanov et al., 2018] for feature inversion, [Singh et al., 2021] proposed *Likelihood Maximization* (LM), which took full advantage of a fixed input Autoencoder network $M_{AE}$ producing $x = M_{AE}(\cdot)$ and replaced the target optimization by minimizing the loss $l_2(M_C(M_{AE}(\cdot)), z^*)$, which significantly improved the optimization-based DRA effect. For learning-based DRA, [He et al., 2019] introduced *Inverse-Network* (IN) that leverages a certain amount of $(z^*, x^*)$ pairs gained by querying the $M_C$ to train a model $M_C^{-1}$ such that $x = M_C^{-1}(z^*)$. Similarly, $l_2$ norm in the pixel space is also used as the loss function. Facing serious threats of existing DRAs, a variety of defense mechanisms have been proposed to greatly mitigate privacy leakage in SI. They are specifically designed to minimize the disclosure of sensitive information from DRA, while still preserving the practical utility of the inference data.

The Model Inversion Attack [Zhang et al., 2020; Chen et al., 2021; An et al., 2022] seeks to extract sensitive features of an individual in the training data, by leveraging the coupled feature information contained in the confidence score of an ID classification model. The Gradient Inversion Attack [Zhu et al., 2019; Geiping et al., 2020; Jeon et al., 2021] aims to recover original training data from shared gradients. Unlike them, DRA focuses on reconstructing private inference data using feature representation output from the split layer of any functional DNN models.

## 2.3 StyleGAN & GAN Inversion

The StyleGAN generator consists of a mapping network $f$ and a synthesis network $G_{Style}$. In a typical image generation process of StyleGAN, a latent vector $z$ is sampled from the $\mathcal{Z}$ space, which follows the Gauss Distribution. Then an intermediate latent vector $w$ is obtained from $f(z)$. The $f$ is a mapping network implemented by an 8-layer Multi-Layer Perceptron (MLP), making the generation based on a disentangled representation. Finally, the $w$ is copied $N$ times ($N = log(output\_size, 2) * 2 - 2$) and leveraged to control layer-grained adaptive instance normalization (AdaIN)[Huang and Belongie, 2017] operations, as shown in Figure 2.

With the rapid development of the StyleGAN series network, a variety of GAN Inversion methods have emerged[Abdal et al., 2019, 2020; Richardson et al., 2021; Wang et al., 2023], which aim to invert a given image back into the latent space of a pre-trained GAN model. Especially for StyleGAN, several latent spaces ($\mathcal{W}+, \mathcal{S}, \mathcal{P}, \mathcal{P}+$)[Zhu et al., 2020b] and formulations (learning, optimization, or both)[Xia et al., 2022] are utilized to achieve better inversion results. Different from common GAN Inversion that focuses on distortion-editability trade-off[Zhu et al., 2020a; Tov et al., 2021], in this paper we customize advanced GAN Inversion technologies to DRA in SI, concentrating on raising the quality of reconstruction.

# 3 Methodology

In this section, we first analyze the threat model of DRA in SI. Then, we formulate the design details and corresponding intuitions of our *GLASS* and *GLASS++*.

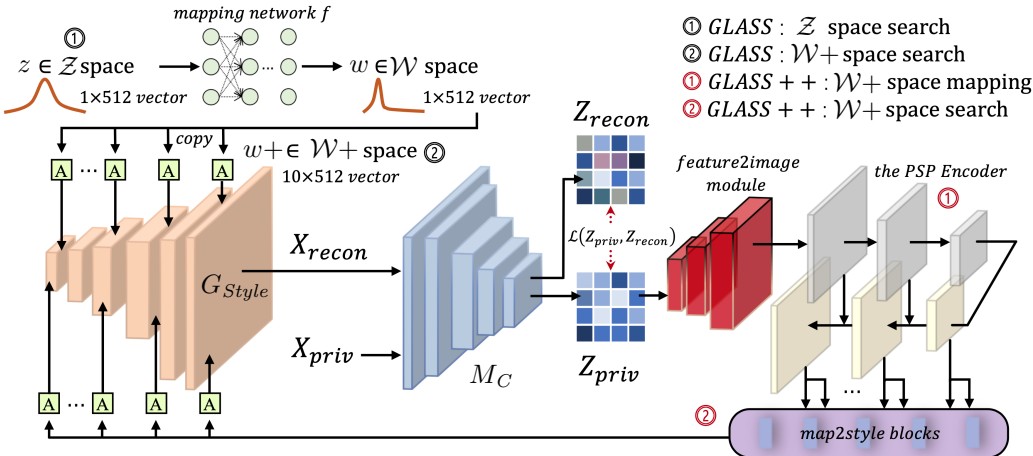

Figure 2: The framework overview of *GLASS* & *GLASS++*.

## 3.1 Threat Model

We assume an honest-but-curious server-side adversary in SI, who receives the client's intermediate feature representations and tries to reconstruct the private inference instances from them. The adversary has the parameters and structure of the whole target model $M_T$, as a white-box setting. This is a reasonable assumption in real-world scenarios, where typically the service provider of SI needs to obtain the target model and perform the splitting setup before deploying it. We also take into account the strictest case that the training of the target model is accompanied by defensive purposes. This usually occurs when the target model is provided to the server by a trusted model provider that sets defense mechanisms, or when the training objectives of the target model are forcibly set by the client with privacy protection requirements. In addition, we extend the consideration to scenarios where the adversary has limited capabilities, i.e., having no client model $M_C$ or being non-queryable, as a black-box setting or a query-free setting. Besides, we assume that the adversary has an auxiliary public dataset $\mathcal{D}_A$ with a similar distribution to that of the target model training dataset. Later in Section 5, we show that our attacks can be carried out effectively, even using an auxiliary dataset with a certain distribution shift.

## 3.2 *GLASS*

**Setup.** Before launching the *GLASS*, a pre-trained StyleGAN generator $G_{Style}$ is necessary, which is trained on a data distribution similar to the private inference data $\mathcal{D}_P$. This is easy to obtain because StyleGAN models pre-trained on various data distributions (especially structured data distributions such as faces) are widely released online. We first formalize Optimization-based DRA as:

$$\min \ \mathcal{L}(M_C(x), z^*) \tag{1}$$

, where $\mathcal{L}$ is the $l_2$-distance between two intermediate features, and the adversary tries to find an $x$ closest to $x^*$ through optimization.

$\mathcal{Z}$ **space search.** The non-convexity of StyleGAN generation makes the optimization problem non-convex. *For the optimization of non-convex functions, an ideal initial point and certain disturbance is extremely significant.* The step ① of *GLASS* is searching in $\mathcal{Z}$ space for the reason that the entanglement of $\mathcal{Z}$ space increases the amplitude of positive perturbation (come from optimizer Adam, SGD, etc) in representation space, which avoids the minimization via gradient descent resulting in poor local minima in the same degree. The formal representation of $\mathcal{Z}$ space search is:

$$\min_{z \in \mathcal{Z}} \ \mathcal{L}(M_C(G_{Style}(z)), M_C(x^*)) + \lambda \, \mathcal{R}(G_{Style}, z) + \alpha \, \mathcal{TV}(G_{Style}(z)) \tag{2}$$

. Obtaining distorted images is a common occurrence by directly searching according to $\mathcal{L}$, as the optimization process may cause $z$ to deviate significantly from the distribution of $\mathcal{Z}$ space. Therefore,

we adopt *KL-based regularization* [Kingma and Welling, 2013] to constrain the optimization process of $z$ to conform to the normal distribution as:

$$\mathcal{R}(G_{Style}, z) = -\frac{1}{2} \sum_{i=1}^{k} (1 + \log(\sigma_i^2) - \mu_i^2 - \sigma_i^2) \tag{3}$$

, where $\mu_i^2$ and $\sigma_i^2$ represent the element-wise mean and standard deviation. The regularization item $\mathcal{R}(\cdot)$ reduces the Kullback-Leibler divergence between $z$ and the standard Gaussian distribution $\mathcal{N}(0, 1)$, controlled by $\lambda$. Furthermore, we adopt *Total Variation* [Rudin et al., 1992] to bring prior information of the natural image, which encourages the generated image $x = G_{Style}(z)$ to be piece-wise smooth, controlled by $\alpha$. Defined as:

$$\mathcal{TV}(x) = \sum_{i,j} \sqrt{|x_{i+1,j} - x_{i,j}|^2 + |x_{i,j+1} - x_{i,j}|^2} \tag{4}$$

$\mathcal{W}+$ **space search.** Based on the $z$ got in step ①, we perform $f(z)$ to obtain $w$, then copy it 10 times as $w+$. In step ②, we search the $\mathcal{W}+$ space as follows:

$$\min_{w+\in\mathcal{W}+} \mathcal{L}(M_C(G_{Style}(w+)), M_C(x^*)) + \alpha\,\mathcal{TV}(G_{Style}(w+)) \tag{5}$$

. For adequate disentanglement of $\mathcal{W}+$ space, it is efficient to find the extreme point in the representation space. The two-step search algorithm is intuitively efficient. As depicted in Figure 3 (A), the search in $\mathcal{Z}$ space during step ① introduces significant perturbations, thereby preventing the initial point $z$ from getting trapped in local optima. Subsequently, leveraging the superior editability provided by the highly disentangled representations in $\mathcal{W}+$ space, the search in step ② further enhances the resemblance between the reconstructed data and the private input data. The iterative optimization process ultimately achieves the global optimum $w+_2$ (the global optimum refers to the optimum attainable by the attacker with existing knowledge). The detailed algorithm of *GLASS* can be found in Appendix A.1.

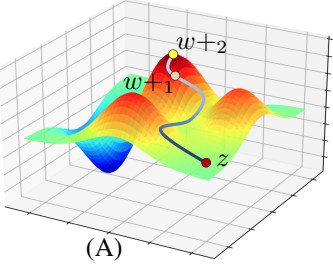 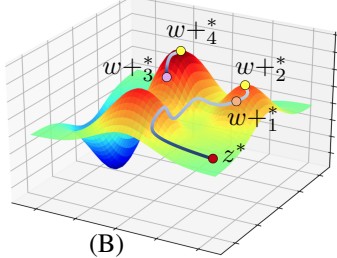

(A)                                      (B)

Figure 3: (A) Through the large disturbance brought by $\mathcal{Z}$ space search, *GLASS* can easily optimize $z$ to $w+_1$ and further reach the global optimum $w+_2$ by $\mathcal{W}+$ space search. (B) When the available information for reconstruction is tiny, $\mathcal{Z}$ space search will inevitably fall into the local optimal point even accompanying disturbance, making *GLASS* optimize $z^*$ to $w+_1^*$ and $\mathcal{W}+$ space search to $w+_2^*$. While *GLASS++* utilizes the mapping relationship between feature space and latent space to obtain an improved initial point $w+_3^*$ and subsequently achieves the global optimal $w+_4^*$.

### 3.3  *GLASS++*

**Intuition.** *GLASS*, this purely optimization-based DRA method could achieve remarkable results at shallow split points. However, when the split point is deep, the $(Width, Height)$ of the intermediate feature representation becomes far less than that of the private input, meaning that the spatial information for reconstruction is gradually transformed into the semantic information needed for the classification task. The less spatial information available makes the optimization process more difficult, causing the results of *GLASS* to fall into pool local optima. As shown in Figure 3 (B), even with a huge disturbance amplitude, $z_2$ eventually falls into a local extreme point $w+_1^*$, due to the lack of information. This results in little improvement in the subsequent $\mathcal{W}+$ space search. Therefore, we propose *GLASS++* to alleviate the above problem.

**Setup.** Before launching the *GLASS++*, we introduce $pixel2style2pixel$ Encoder $E_{PSP}$ and *map2styles* blocks $M_m$ from [Richardson et al., 2021]. Tensors of the image space are fed to $E_{PSP}$ to generate three levels of feature maps. These feature maps are subsequently utilized by the *map2styles* blocks $M_m$ to extract desired styles $w+$. We incorporate these technologies into our DRA framework. For more details on these components, please refer to Appendix B.1.

$\mathcal{W}+$ **space mapping & search.** To tackle the problem of non-convex function optimization tending to fall into local optima, we employ Learning-based methods. These methods involve mapping the intermediate feature representation to $\mathcal{W}+$ space, resulting in a more advantageous initial point. We specifically design a *feature2image* module $M_f$, projecting vectors from high-dimensional feature space into image space. This module serves as a valuable tool for facilitating subsequent style extraction. Then we joint the $E_{PSP}$ and $M_m$ to extract desired styles $w+$. In step ①, these three items collaboratively perform as Encoder $E = M_f \circ E_{PSP} \circ M_m$, mapping the vectors from feature space to $\mathcal{W}+$ space as:

$$w+ = E(M_C(x^*)) \tag{6}$$

, with the help of $E$, a point near the global optimum is determined as the initial point of $\mathcal{W}+$ space search in step ②. As shown in Figure 3 (B), it is considerably more effortless to reach $w+_4^*$ from $w+_3^*$ driven by gradient descent. The detailed algorithm of *GLASS++* can be found in Appendix A.2.

### 3.4 Approach Analysis

Referring to [Jeon et al., 2021], we formalize the DRA under SI as an optimization problem. Through a pre-trained StyleGAN generator, the problem of (1) can be better solved by transferring from $\mathbb{R}^m$ to $\{G_{Style}(l) : l \in \mathbb{R}^k\}$, where $l$ is a latent code in either $\mathcal{Z}$ or $\mathcal{W}+$ space, $k$ denotes the dimension of $l$, and $m$ refers to the dimension of image space. Hence, *GLASS* and *GLASS++* perform the latent spaces search as follows:

$$\min_{l \in \mathbb{R}^k} \mathcal{D}(M_C(G_{Style}(l)), M_C(x^*)) \tag{7}$$

, where $\mathcal{D}$ represents total loss terms in latent spaces search. When the private inference data is approximated with a sufficient narrow error, the DRA through latent spaces search in (7) aligns with image space search in (1).

## 4 Evaluation

We systematically evaluate our proposed attacks in terms of their performance against representative image classification tasks and compare them with existing attack methods. Additionally, we measure the robustness of seven defense mechanisms against various DRAs. We implement *GLASS* and *GLASS++* in Pytorch[Paszke et al., 2019]. Most experiments are carried out on a server equipped with 256 GB RAM, two Intel Xeon Gold 6133, and four NVIDIA RTX 4090 GPUs.

### 4.1 Experimental Settings

**Datasets & Tasks.** we use (1) CelebA[Liu et al., 2015] containing 202,599 face images of 10,177 identities, (2) FFHQ[Karras et al., 2019] containing 70,000 face images with considerable variation in terms of age, ethnicity and image background. Both are scaled down to $64 \times 64$ pixels. We study DRA against models built for the *Attractiveness Classification* task: Binary attractiveness classification performed on the CelebA, as we consider attractiveness to be a remarkably inclusive facial attribute. We adopt ResNet-18[He et al., 2016] and split the target model $M_C$ into different layers, as shown in Appendix B.2.

**Attack Setup.** We split the datasets into two parts: a private dataset $\mathcal{D}_P$ for training the target model and a public data used as an auxiliary dataset $\mathcal{D}_A$ for training our StyleGAN model. For CelebA, we selected 80,525 images belonging to 3,000 identities with the highest number of images as $\mathcal{D}_P$, while the remaining images from other identities as public data $\mathcal{D}_A$. This scheme ensures that there are ***no overlapping identities*** between $\mathcal{D}_A$ and $\mathcal{D}_P$ in all experiments. This means that the public data only helps the adversary obtain general information about the features as prior knowledge,

without providing any class-specific information relevant to the private data[Chen et al., 2021]. For practical reasons, we utilize the FFHQ dataset as public data to train our StyleGAN model, as there are numerous StyleGAN models available on the Internet that are trained with FFHQ. Note that there is a certain distribution shift between the FFHQ dataset and the celeba dataset[Kahla et al., 2022]. To be fair, We set the number of iterations for Optimization-based DRA to 20,000 and the number of training epochs for Learning-based DRA to 30, while incorporating *Total Variation* into each attack loss function. It is worth acknowledging that the influence of hyperparameters varies across different adversarial settings and defense mechanisms. We analyze the hyperparameter selection strategies within different settings to meet the reasonable effectiveness of various attacks. Detailed information regarding the hyperparameters can be found in Appendix B.3.

**Compared Baselines & Evaluation Protocol.** We set the three existing DRAs as the baseline attacks: *regularized Maximum Likelihood Estimation* (rMLE)[He et al., 2019], *Likelihood Maximization* (LM)[Singh et al., 2021] and *Inverse-Network* (IN)[He et al., 2019]. For the sake of generality, our experiments are conducted on 40 randomly selected and fixed images, and the mean value of each evaluation metric is calculated as the result.

**Evaluation Metrics.** In addition to visually quantifying reconstruction attacks, we selected five metrics to evaluate the similarity between the original image and the reconstructed image: Learned Perceptual Image Patch Similarity (LPIPS ↓)[Zhang et al., 2018], Structural Similarity Index (SSIM ↑)[Wang et al., 2004], Peak Signal-to-Noise Ratio (PSNR ↑)[Hore and Ziou, 2010], Mean Squared Error (MSE ↓) and Natural Image Quality Evaluator (NIQE ↓)[Mittal et al., 2012]. Note that "↓" means the lower the metric the higher the relative image quality, while "↑" represents the higher the metric the higher the image quality.

## 4.2 Attack Performance

Figure 4 shows the reconstruction attack performance of our methods. In general, our methods demonstrate optimal and second-best performance across nearly all split points, whether compared with Optimization-based or Learning-based DRAs. Specifically, when the split point is set to Block3, our Optimization-based *GLASS* reduces LPIPS from **0.298** to **0.140** compared to LM; Our Learning-based *GLASS++* enhances the SSIM from **0.685** to **0.833** compared to IN (the full results are in Appendix C). When the split point is located at a deeper Block5, *GLASS* exhibits a remarkable ability in reconstructing faces with features highly similar to the ground truth. In contrast, other Optimization-based DRAs fail to reveal any valid sensitive information. We attribute this superior performance to two main factors: the wealth of prior knowledge embedded in the pre-trained StyleGAN model and the powerful feature search capability brought by our methods. Note that LM achieves optimal NIQE on Block5, but we observe that its reconstructed images are almost noisy. We analyze that this was due to the instability of NIQE on small-size images, which resulted in extremely poor attack results but good values. Therefore, we exclude NIQE from the experiments in subsection 4.3.

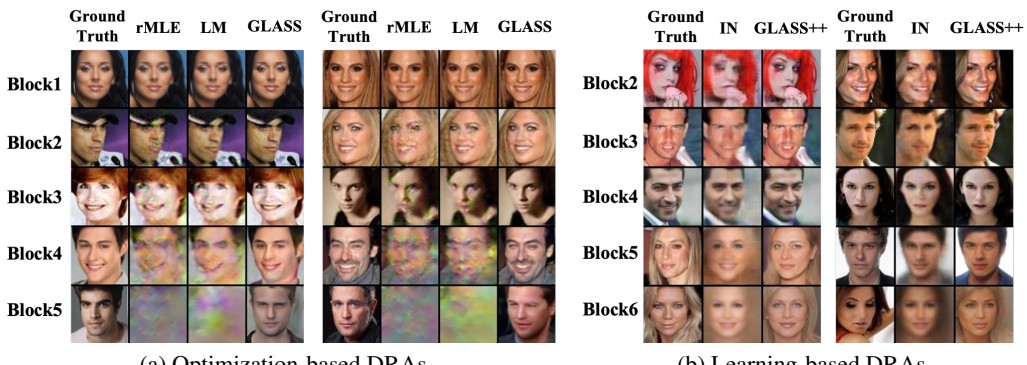

(a) Optimization-based DRAs            (b) Learning-based DRAs

Figure 4: White-Box DRAs against the original model under different split points.

## 4.3 Defense Mechanisms Assessment

**Reconstruction Performance under Defense Mechanisms.** The quantitative metrics in Table 1 demonstrate the remarkable effectiveness of our methods in breaching all defenses, surpassing the performance of other baseline attacks even when the accuracy of the target model declines due to defense. Our *GLASS++* and *GLASS* consistently yield the optimal and the second-best results on most of the metrics. Even against state-of-the-art defense mechanisms like NoPeek, Adversarial Learning, and DISCO, we can still carry out powerful DRA, as shown in Figure 5. Specifically, when against NoPeek (Acc=79.97%), *GLASS* and *GLASS++* increased the SSIM of reconstructed images to **0.554** and **0.678**, compared to **0.006**, **0.233** and **0.376** of baseline attacks, showcasing that our methods turn the failure into the successful attack. It is crucial to emphasize that even a **3%** loss in accuracy is unsatisfactory because the target task is a simple binary classification problem. However, in practice, DL tasks are often very complex, and even a slight privacy defense may lead to poor performance.

Table 1: According to the attack effect of IN, we adjust the hyperparameters of different defenses to provide the defended model with three levels of privacy protection, resulting in a gradual decrease in model accuracy (original Acc=79.97 %). All experiments are performed on Block3. Implementation details and hyperparameters of defenses are in Appendix B.4.

| Defense | Method | Acc = 79.97% | | | | Acc = 79.23% | | | | Acc = 78.56% | | | |
|---|---|---|---|---|---|---|---|---|---|---|---|---|---|
| | | LPIPS↓ | SSIM↑ | PSNR↑ | MSE↓ | LPIPS↓ | SSIM↑ | PSNR↑ | MSE↓ | LPIPS↓ | SSIM↑ | PSNR↑ | MSE↓ |
| NoPeek | rMLE | 0.853 | 0.006 | 7.059 | 1.048 | 0.855 | 0.006 | 7.099 | 1.039 | 0.856 | 0.006 | 7.105 | 1.038 |
| | LM | 0.679 | 0.233 | 11.74 | 0.449 | 0.729 | 0.171 | 11.00 | 0.500 | 0.766 | 0.178 | 10.29 | 0.539 |
| | IN | 0.638 | 0.376 | 15.57 | 0.160 | 0.607 | 0.377 | **15.51** | **0.156** | 0.618 | **0.352** | **15.15** | **0.168** |
| | **GLASS** | 0.293 | 0.554 | 17.53 | 0.142 | 0.351 | 0.422 | 14.68 | 0.244 | 0.414 | 0.343 | 13.46 | 0.326 |
| | **GLASS++** | 0.220 | 0.678 | 21.60 | 0.070 | 0.295 | 0.549 | 18.04 | 0.160 | 0.301 | 0.532 | 18.21 | 0.115 |

| Defense | Method | Acc = 79.34% | | | | Acc = 79.26% | | | | Acc = 79.04% | | | |
|---|---|---|---|---|---|---|---|---|---|---|---|---|---|
| | | LPIPS↓ | SSIM↑ | PSNR↑ | MSE↓ | LPIPS↓ | SSIM↑ | PSNR↑ | MSE↓ | LPIPS↓ | SSIM↑ | PSNR↑ | MSE↓ |
| DISCO | rMLE | 0.618 | 0.330 | 14.69 | 0.228 | 0.784 | 0.147 | 11.87 | 0.367 | 0.763 | 0.162 | 11.96 | 0.361 |
| | LM | 0.387 | 0.620 | 19.96 | 0.127 | 0.762 | 0.220 | 12.48 | 0.330 | 0.799 | 0.205 | 11.48 | 0.422 |
| | IN | 0.410 | 0.584 | 20.59 | 0.049 | 0.626 | 0.336 | 15.84 | **0.149** | 0.636 | 0.314 | **15.56** | **0.158** |
| | **GLASS** | 0.165 | 0.776 | 24.76 | 0.022 | 0.328 | 0.464 | 16.37 | 0.154 | 0.345 | 0.431 | 15.55 | 0.201 |
| | **GLASS++** | 0.152 | 0.784 | 24.75 | 0.021 | 0.241 | 0.584 | 18.79 | 0.082 | 0.260 | 0.576 | 18.46 | 0.086 |

| Defense | Method | Acc = 79.13% | | | | Acc = 78.76% | | | | Acc = 78.13% | | | |
|---|---|---|---|---|---|---|---|---|---|---|---|---|---|
| | | LPIPS↓ | SSIM↑ | PSNR↑ | MSE↓ | LPIPS↓ | SSIM↑ | PSNR↑ | MSE↓ | LPIPS↓ | SSIM↑ | PSNR↑ | MSE↓ |
| Adv-Learning | rMLE | 0.591 | 0.327 | 14.60 | 0.252 | 0.654 | 0.200 | 12.69 | 0.343 | 0.678 | 0.155 | 12.03 | 0.381 |
| | LM | 0.308 | 0.667 | 20.80 | 0.139 | 0.414 | 0.533 | 17.19 | 0.217 | 0.480 | 0.427 | 15.04 | 0.258 |
| | IN | 0.483 | 0.517 | 19.82 | **0.059** | 0.584 | 0.382 | 17.57 | **0.100** | 0.601 | 0.342 | 16.45 | 0.130 |
| | **GLASS** | 0.155 | 0.765 | 23.68 | 0.067 | 0.208 | 0.626 | 18.26 | 0.148 | 0.197 | 0.686 | 20.17 | 0.110 |
| | **GLASS++** | 0.131 | 0.816 | 25.93 | 0.016 | 0.142 | 0.794 | 24.69 | 0.023 | 0.157 | 0.763 | 23.21 | 0.040 |

| Defense | Method | Acc = 78.57% | | | | Acc = 78.04% | | | | Acc = 77.83% | | | |
|---|---|---|---|---|---|---|---|---|---|---|---|---|---|
| | | LPIPS↓ | SSIM↑ | PSNR↑ | MSE↓ | LPIPS↓ | SSIM↑ | PSNR↑ | MSE↓ | LPIPS↓ | SSIM↑ | PSNR↑ | MSE↓ |
| Noise Mask | rMLE | 0.703 | 0.227 | 14.65 | 0.188 | 0.771 | 0.099 | 11.58 | 0.378 | 0.835 | 0.021 | 7.90 | 0.869 |
| | LM | 0.612 | 0.409 | 18.30 | 0.079 | 0.708 | 0.225 | 14.61 | 0.185 | 0.791 | 0.057 | 8.78 | 0.709 |
| | IN | 0.352 | 0.610 | 21.20 | 0.042 | 0.424 | 0.549 | 19.71 | 0.059 | 0.590 | **0.424** | **16.22** | **0.133** |
| | **GLASS** | 0.230 | 0.660 | 21.66 | 0.039 | 0.245 | 0.607 | 20.16 | 0.056 | 0.360 | 0.396 | 15.18 | 0.174 |
| | **GLASS++** | 0.184 | 0.714 | 22.57 | 0.032 | 0.231 | 0.626 | 20.52 | 0.051 | 0.283 | 0.524 | 17.71 | 0.096 |

| Defense | Method | Acc = 78.09% | | | | Acc = 77.65% | | | | Acc = 77.35% | | | |
|---|---|---|---|---|---|---|---|---|---|---|---|---|---|
| | | LPIPS↓ | SSIM↑ | PSNR↑ | MSE↓ | LPIPS↓ | SSIM↑ | PSNR↑ | MSE↓ | LPIPS↓ | SSIM↑ | PSNR↑ | MSE↓ |
| Dropout Defense | rMLE | 0.455 | 0.563 | 20.67 | 0.048 | 0.539 | 0.522 | 19.58 | 0.062 | 0.610 | 0.426 | 17.29 | 0.110 |
| | LM | 0.319 | 0.717 | 23.44 | 0.026 | 0.408 | 0.659 | 22.21 | 0.034 | 0.483 | 0.606 | 21.11 | 0.044 |
| | IN | 0.350 | 0.612 | 21.12 | 0.043 | 0.425 | 0.555 | 20.21 | 0.053 | 0.499 | 0.490 | 18.83 | 0.073 |
| | **GLASS** | 0.144 | 0.790 | 25.31 | 0.018 | 0.171 | 0.766 | 24.23 | 0.023 | 0.195 | 0.712 | 22.83 | 0.031 |
| | **GLASS++** | 0.144 | 0.797 | 25.30 | 0.018 | 0.163 | 0.777 | 24.46 | 0.022 | 0.169 | 0.766 | 24.23 | 0.022 |

| Defense | Method | Acc = 79.23% | | | | Acc = 78.65% | | | | Acc = 77.73% | | | |
|---|---|---|---|---|---|---|---|---|---|---|---|---|---|
| | | LPIPS↓ | SSIM↑ | PSNR↑ | MSE↓ | LPIPS↓ | SSIM↑ | PSNR↑ | MSE↓ | LPIPS↓ | SSIM↑ | PSNR↑ | MSE↓ |
| Shredder | rMLE | 0.680 | 0.231 | 14.38 | 0.217 | 0.700 | 0.200 | 13.67 | 0.250 | 0.676 | 0.234 | 14.37 | 0.220 |
| | LM | 0.547 | 0.475 | 19.09 | 0.074 | 0.564 | 0.448 | 18.44 | 0.086 | 0.538 | 0.481 | 18.97 | 0.072 |
| | IN | 0.373 | 0.604 | 21.03 | 0.044 | 0.383 | 0.602 | 20.94 | **0.044** | 0.390 | 0.596 | 20.87 | **0.045** |
| | **GLASS** | 0.222 | 0.668 | 21.82 | 0.042 | 0.246 | 0.634 | 20.95 | 0.047 | 0.236 | 0.646 | 21.32 | 0.045 |
| | **GLASS++** | 0.207 | 0.695 | 22.27 | 0.034 | 0.213 | 0.695 | 22.19 | 0.035 | 0.211 | 0.697 | 22.06 | 0.036 |

| Defense | Method | Acc = 78.74% | | | | Acc = 78.44% | | | | Acc = 77.56% | | | |
|---|---|---|---|---|---|---|---|---|---|---|---|---|---|
| | | LPIPS↓ | SSIM↑ | PSNR↑ | MSE↓ | LPIPS↓ | SSIM↑ | PSNR↑ | MSE↓ | LPIPS↓ | SSIM↑ | PSNR↑ | MSE↓ |
| Siamese Defense | rMLE | 0.745 | 0.155 | 12.78 | 0.304 | 0.740 | 0.176 | 12.74 | 0.308 | 0.746 | 0.175 | 12.48 | 0.324 |
| | LM | 0.574 | 0.451 | 16.97 | 0.157 | 0.604 | 0.405 | 16.40 | 0.176 | 0.670 | 0.335 | 14.67 | 0.228 |
| | IN | 0.378 | 0.614 | 21.29 | 0.041 | 0.386 | 0.592 | 21.00 | 0.044 | 0.425 | 0.553 | 20.28 | 0.052 |
| | **GLASS** | 0.166 | 0.797 | 25.30 | 0.023 | 0.186 | 0.751 | 23.77 | 0.033 | 0.201 | 0.731 | 23.07 | 0.046 |
| | **GLASS++** | 0.144 | 0.826 | 26.30 | 0.016 | 0.159 | 0.802 | 25.33 | 0.022 | 0.159 | 0.797 | 25.14 | 0.021 |

**Defense Analysis.** For Clipping type defenses (Dropout Defense, DISCO), a high clipping rate is essential for providing sufficient privacy protection (about 90% intermediate feature clipping or 95% channel clipping). Dropout Defense is difficult to defend against Optimized-based DRAs because the Mask (randomly set to zero) superimposed on the intermediate feature representation can be easily picked up. DISCO's superior defense effect comes from 95% channel clipping, which is possible

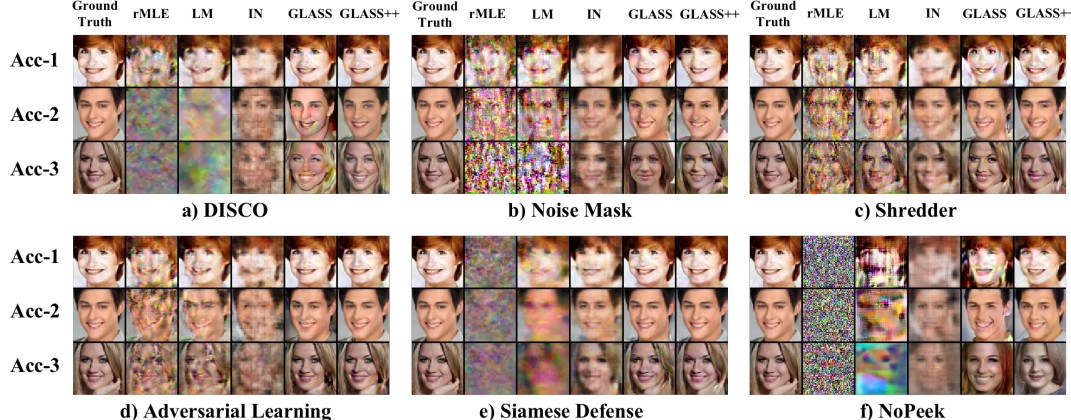

Figure 5: The results of different defenses against DRAs. The accuracy rate corresponds to the Acc in Table 1, as Acc-1>Acc-2>Acc-3.

because the target task is relatively simple, resulting in highly redundant feature channels. For Noise Addition type defenses (Noise Mask, Shredder), Optimization-based DRAs are noise-sensitive. Even in a White-box setting, random sampling noise can still greatly interfere with the optimization process, leading to poor attack effects. In contrast, Learning-based DRAs demonstrate a certain robustness to noise. For further analysis of Shredder, we collect intermediate feature representations superimposed with Shredder noise and apply t-SNE[Van der Maaten and Hinton, 2008] to reduce their dimensionality. Notably, even when sampling from **50** noise distributions, the intermediate features can be distinctly clustered into 50 groups, as shown in Appendix F. We can degrade the noise distribution library of Shredder (default is 20) to a single distribution by using carefully selected intermediate feature representations (which can be mapped to the corresponding target group) during the optimization process, which greatly improves the effect of DRA. For Feature Obfuscating type defenses (Adversarial Learning, NoPeek, Siamese Defense), we believe that they only serve to increase the difficulty of feature matching in DRA. However, with a sufficiently powerful feature search capability, the attack is easy to implement. Appendix D.1 illustrates the curves of feature loss between the model with NoPeek and the original model under the *GLASS*, targeting the same image during optimization. It is evident that when the optimization process converges, the numerical difference between them is two orders of magnitude. In Appendix D.2, we establish a correlation between the model's accuracy and the SSIM of DRA under different attacks and defenses. Notably, NoPeek and DISCO emerge as the most effective defense mechanisms. As shown in Appendix D.3(a), the SSIM of our attacks is mostly above 0.5, and the slope of the broken lines is large, which means that our attacks are more robust.

## 5 Extended Experiments

**Black-box & Query-free Settings.** We relax the assumption of adversary capability and use gradient-free optimization technology to implement Black-box DRA. For Query-free, we adapt model fine-tuning and $\mathcal{P}+$ space cropping. The experimental results show that the gradient-free optimization only takes 2,000 iterations to obtain effective information, and the customized *GLASS* improves the attack SSIM of Block2 from 0.152 to 0.392 in the Query-free setting. Experimental details and results are in Appendix E.1

**Heterogeneous Data.** We further extend the reconstruction attack to heterogeneous data like CINIC-10 [Darlow et al., 2018]. We chose to utilize the gradient-free CMA optimizer because CINIC-10 has a more heterogeneous data distribution than well-aligned datasets like CelebA, making it hard for the gradient-based optimizer to search in latent space[Li et al., 2022]. In our implementation, we utilize a publicly released StyleGAN-XL model trained on CIFAR-10 [Krizhevsky et al., 2009] to construct *GLASS*. Figure 6 shows part of the results. It can be seen that owing to rich prior knowledge, our *GLASS* can obtain highly similar semantic information and generate more vivid images than IN. Experimental details are in Appendix E.2.

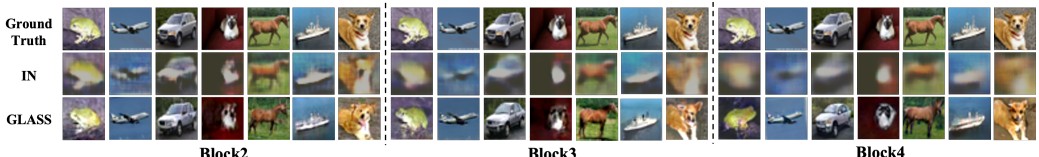

Figure 6: DRAs against Heterogeneous Data.

**Data Distribution Shift.** We replaced the StyleGAN model used in the attack from the one trained on the CelebA to the one trained on the FFHQ. Table 2 shows that *GLASS++* is still very effective, even if the prior knowledge is learned from a data distribution that differs from the private data distribution (facial structure alignment and feature diversity difference). Furthermore, we enhance the robustness of the attack by incorporating a fundamental domain adaptation technique, specifically model fine-tuning. We concatenate a step of model parameter optimization after *GLASS++* to make the generator model parameters trainable, which further aligns the features of the reconstructed images with the target features.

Table 2: Reconstruction attacks performance of Data Distribution Shift on Block3.

| Dataset | Method | Evaluation Metrics | | | | |
|---------|--------|----------|--------|--------|-------|--------|
| | | LPIPS ↓ | SSIM ↑ | PSNR ↑ | MSE ↓ | NIQE ↓ |
| CelebA | GLASS++ | 0.128 | 0.833 | 26.56 | 0.014 | 15.30 |
| FFHQ | GLASS++ | 0.211 | 0.727 | 23.95 | 0.025 | 13.77 |
| | GLASS++ Fine-tuning | 0.199 | 0.736 | 24.22 | 0.023 | 14.75 |

Moreover, we carry out a more practical implementation of DRA on uncropped/unaligned private inference data to enhance the quality of our work. According to [Yang et al., 2023], we enhance StyleGAN by transitioning its constant first-layer feature to a variable one. We integrate this with the latent code of $\mathcal{W}+$ space and undertake joint optimization during the second stage of our methodology. As demonstrated in Figure 7, the evaluation of *GLASS* underscores our method's effectiveness even when dealing with transformed or natural privacy inference data.

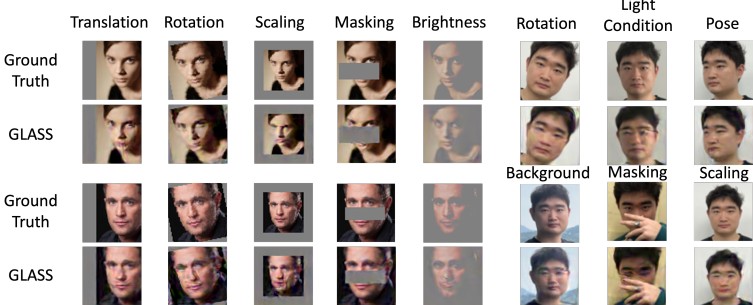

Figure 7: The results of GLASS when private inference data are transformed or natural images. The split point is set to Block3.

# 6 Conclusion

In this paper, we propose *GLASS* and *GLASS++*, the enhanced DRAs against SI. Our experiments demonstrate the effectiveness of our attacks on different split points and various adversarial settings. We anticipate that our proposed attacks will spotlight the significance of safeguarding privacy in split inference systems and encourage the advancement of more robust defense mechanisms. Regarding the limitation, it mainly comes from the inherent flaw of Optimization-based attacks, for single image optimization is less efficient than Learning-based attacks.

**Acknowledgments and Disclosure of Funding** We thank the anonymous reviewers for their constructive comments. This work was supported in part by the National Natural Science Foundation of China under Grants No. 61872430, 61402342, and 61772384 and was sponsored by Ant Group.

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
