# Appendix

## A  Algorithm details

### A.1  GLASS

---

**Algorithm 1** GAN-based latent space search attack (*GLASS*)

---

**Require:** the client model $M_C$; the target intermediate feature representation $z^* = M_C(x^*)$; a pretrained StyleGAN model $G_{Style}$ and mapping network $f$; $l_2$-distance $\mathcal{L}$; *KL-based regularization* coefficient $\lambda$; $\mathcal{TV}$ *regularization* coefficient $\alpha$;

1: Initialize $z \in \mathcal{Z}$ space randomly
2: Find $z \leftarrow \arg\min_z \ \mathcal{L}(M_C(G_{Style}(z)), M_C(x^*)) + \lambda\, \mathcal{R}(G_{Style}, z) + \alpha\, \mathcal{TV}(G_{Style}(z))$
$\qquad\qquad\qquad\qquad\qquad\qquad\qquad\qquad\qquad\qquad\qquad\qquad$ //$\mathcal{Z}$ space search
3: set $w := f(z)$, then copy it 10 times as $w+ \in \mathcal{W}+$ space
4: Find $w+ \leftarrow \arg\min_{w+} \ \mathcal{L}(M_C(G_{Style}(w+)), M_C(x^*)) + \alpha\, \mathcal{TV}(G_{Style}(w+))$
$\qquad\qquad\qquad\qquad\qquad\qquad\qquad\qquad\qquad\qquad\qquad\qquad$ //$\mathcal{W}+$ space search
5: **return** $G_{Style}(w+)$

---

### A.2  GLASS++

---

**Algorithm 2** GAN-based latent space search attack plus plus (*GLASS++*)

---

**Require:** the client model $M_C$; the target intermediate feature representation $z^* = M_C(x^*)$; a pretrained StyleGAN model $G_{Style}$ and mapping network $f$; an Encoder $E = M_f \circ E_{PSP} \circ M_m$; public data $\mathcal{X}$; training epoch $T$; batch size $k$; step size $\epsilon$; $l_2$-distance $\mathcal{L}$; $\mathcal{TV}$ *regularization* coefficient $\alpha$; $l_2$ *regularization* coefficient $\alpha_1$; *LPIPS regularization* coefficient $\alpha_2$; *norm regularization* coefficient $\alpha_3$;

1: $E = EncoderTraining(M_C, \mathcal{X})$
2: Initialize $w+ \in \mathcal{W}+$ space as $w+ := E(M_C(x^*))$
$\qquad\qquad\qquad\qquad\qquad\qquad\qquad\qquad\qquad\qquad\qquad\qquad$ //$\mathcal{W}+$ space mapping
3: Find $w+ \leftarrow \arg\min_{w+} \ \mathcal{L}(M_C(G_{Style}(w+)), M_C(x^*)) + \alpha\, \mathcal{TV}(G_{Style}(w+))$
$\qquad\qquad\qquad\qquad\qquad\qquad\qquad\qquad\qquad\qquad\qquad\qquad$ //$\mathcal{W}+$ space search
4: **return** $G_{Style}(w+)$
5:
6: **Function** $EncoderTraining(M_C, \mathcal{X})$
7: **while** $t < T$ **do**
8: $\quad$ Random sample $x_1, x_2, ..., x_k$ from $\mathcal{X}$
9: $\quad L(E^{(t)}) = \frac{1}{k}\sum_{i=1}^{k}(\alpha_1 \mathcal{L}(M_C(G_{Style}(E(M_C(x_i)))), M_C(x_i))$
10: $\qquad\qquad + \alpha_2 LPIPS(G_{Style}(E(M_C(x_i))), x_i)$
11: $\qquad\qquad + \alpha_3\, norm(E(M_C(x_i))))$
12: $\quad E^{(t+1)} = E^{(t)} - \epsilon * \frac{\partial L(E^{(t)})}{\partial E^{(t)}}$
13: $\quad t += 1$
14: $\quad$ **return** $E^{(t)}$
15: **end while**

---

## B  Experimental details

### B.1  Components

*feature2image* **module** $M_f$ $\quad$ $M_f$ consists of multiple groups of deconvolutional layers and batch normalization layers, projecting vectors from high-dimensional feature space into image space. Each additional group doubles the $(Width, Height)$ of the feature representations.

*pixel2style2pixel* **Encoder** $E_{PSP}$ **and** *map2styles* **blocks** $M_m$     We utilize the official implementation of $E_{PSP}$ and $M_m$ from [Richardson et al., 2021][2]. $E_{PSP}$ is a standard feature pyramid over a ResNet backbone. $M_m$ is a fully convolutional network consisting of 2-strided convolutions and LeakyReLU activations. For each of the target styles, the $M_m$ blocks are trained to extract styles from the corresponding feature maps at three different granularities (small, medium, largest). To adapt to $64 \times 64$-pixel images as input, we modify $E_{PSP}$ and $M_m$ appropriately.

## B.2   Split Points

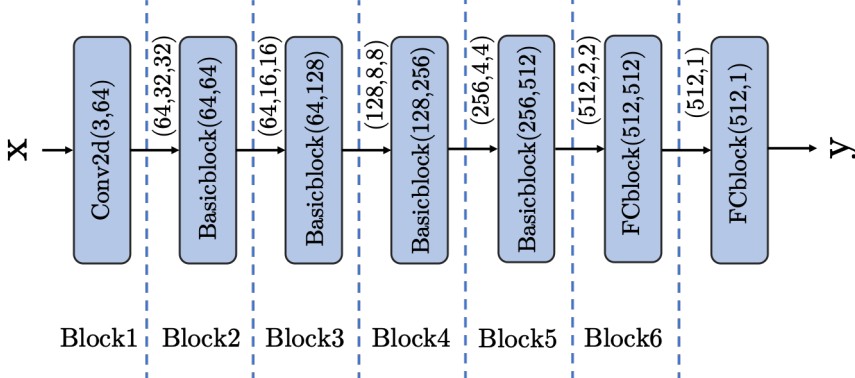

Figure 8: Split Points. The intermediate feature representations include outputs from the convolutional blocks and a fully connected block.

A standard ResNet-18 network is divided into blocks, as shown in Figure 8. From Block1 to Block6, the $(Width, Height)$ of features decreases gradually, which means that spatial information is gradually transformed into semantic information that helps with classification. It is worth noting that we evaluate the scenario where the split point is deep enough (FCblock), making it very challenging for the adversary to carry out DRA.

## B.3   Hyperparameters of DRAs

We configure the following parameters for the Optimization-based rMEL and LM: a learning rate of 1e-2, 20,000 iterations, and $\mathcal{TV}$ *regularization* coefficients of 2 and 1.5, respectively. Similarly, for *GLASS*, we set the learning rate to 1e-2 and the number of iterations to 20,000. This setup ensures fairness and consistency with other Optimization-based DRAs, as we believe that the attack effectiveness of Optimization-based DRAs can be positively influenced by the learning rate and the number of iterations.

Regarding IN, we selected a learning rate of 1e-3 and performed 30 training epochs. For the encoder training of *GLASS++*, we use a learning rate of 1e-2 and trained for 30 epochs. In the subsequent optimization process, we utilize a learning rate of 1e-2 and optimize 8,000 iterations. We set the $\mathcal{TV}$ *regularization* coefficient for our

---

[2]The official implementation of [Richardson et al., 2021]: `https://github.com/eladrich/pixel2style2pixel`

DRAs to 0.01, as the prior knowledge within the StyleGAN model already provides satisfactory regularization.

During the defense examination, we set the $\mathcal{TV}$ *regularization* coefficient for rMEL and LM to 0.3 and make adjustments to the hyperparameters for certain defenses. For example, on NoPeek, we focus the loss solely on the $l_2$-distance between features.

### B.4  Implementation & Hyperparameters of Defenses

**Implementation of Defenses**   Clipping-based Defenses. [He et al., 2020] introduces Dropout Defense, generating random Masks $M$ to apply pixel-level pruning directly to intermediate feature representations as

$$z_{Dropout}^* = M_C(x) \otimes M \tag{8}$$

, each element of $M$ is randomly allocated as 0 with probability $r$ and 1 with probability $1-r$. [Singh et al., 2021] proposes DISCO, using an adversarial network $M_A$ to train a Dynamic Channel Pruning layer $P$, giving consideration to both privacy and utility as

$$L_{util} \triangleq E[l_u(M_S(P(M_C(x^*))), y^*)], \tag{9}$$

$$L_{priv} \triangleq E[l_a(M_A(P(M_C(x^*))), x^*)], \tag{10}$$

$$\min_P [\max_{M_A} -L_{priv} + \rho \min_{M_C, M_S} L_{util}] \tag{11}$$

, where $\rho$ is a hyperparameter to trade-off between utility and privacy. In the inference phase, $P$ clips intermediate feature representations on channel-level according to a feature map score predicted by $P$ for protecting sensitive information in latent representation, as

$$z_{DISCO}^* = P(M_C(x^*), R) \tag{12}$$

. The value of pruning ratio $R$ controls the proportion of channels to be pruned, which provides a trade-off between privacy and utility. We utilize the official implementation of [Singh et al., 2021][3].

Noise Addition-based Defenses. [Titcombe et al., 2021] proposes Noise Mask, which adds additive Laplacian noise to intermediate feature representations as

$$z_{Noise}^* = M_C(x^*) + \varepsilon \tag{13}$$

, bringing disturbance to the sensitive information. When performing inference, a random noise $\varepsilon$ is sampled from a Laplacian distribution parameterized by location $a$ and scale $b$. [Mireshghallah et al., 2020] proposes Shredder, which firstly trains noise tensors overlaid on intermediate feature representations for maintaining utility purposes, then fits each noise tensor to a Laplacian distribution. Finally, the fitted distributions and orders of noise elements are collected. The optimization objective for the noise tensor $T_{noise}$ is as follows:

$$L_{util} \triangleq l_u(M_S(M_C(x^*)), y^*), \tag{14}$$

---

[3]The official implementation of [Singh et al., 2021]: `https://github.com/splitlearning/InferenceBenchmark`

$$L_{priv} \triangleq \texttt{norm}(M_C(x^*) + T_{noise}), \tag{15}$$

$$\min_{T_{noise}} [L_{util} - \alpha \, L_{priv}] \tag{16}$$

. For a noisy inference, Shredder picks one of the collected distributions and stochastically samples a noise tensor from this distribution. Then elements of the noise tensor are rearranged to match the saved order and simply added to the intermediate feature representations.

Feature Obfuscating-based defenses. [Li et al., 2021] adds an additional module Adversary Reconstructer $M_{AR}$ and formalizes the training process as a min-max game (Adversarial Learning):

$$L_{util} \triangleq E[l_u(M_S(M_C(x^*)), y^*)], \tag{17}$$

$$L_{priv1} \triangleq E[1 - \texttt{SSIM}(M_{AR}(M_C(x^*)), x^*)], \tag{18}$$

$$L_{priv2} \triangleq E[1 - \texttt{SSIM}(M_{AR}(M_C(x^*)), I_{noise})], \tag{19}$$

$$\min_{M_C}[\max_{M_{AR}} -L_{priv1} + L_{priv2} + \min_{M_S} L_{util}] \tag{20}$$

, where the $I_{noise}$ is one additional Gaussian noise image. The target is maintaining the utility of the task while transforming the intermediate feature representations so that there is less sensitive information that can be explored by $M_{AR}$. Specifically, the trade-off is controlled by $\lambda_1$ when the $M_C$ is optimized as

$$M_C = \arg\min_{M_C}[L_{util} + \lambda_1(L_{priv2} - L_{priv1})] \tag{21}$$

. [Vepakomma et al., 2020] introduces NoPeek, adapting distance correlation minimization to the training process. For decreasing the distance correlation[4] between input and intermediate feature representations, NoPeek enables the client model to reduce redundant sensitive information of intermediate feature representations from raw input, expressed as

$$L_{util} \triangleq E[l_u(M_S(M_C(x^*)), y^*)], \tag{22}$$

$$L_{priv} \triangleq E[\texttt{DCOR}(M_C(x^*), x^*)], \tag{23}$$

$$\min_{M_C, M_S} [L_{util} + \lambda_2 \, L_{priv}] \tag{24}$$

, where $\lambda_2$ controls the trade-off between utility and privacy. Similar to NoPeek, [Osia et al., 2020] fine-turning the original model with Siamese architecture as an additional loss item to make the representation of the same labeled points closer to each other, while the representation of dissimilar points falls far from each other. This can be expressed as

$$L_{Siamese} = \begin{cases} \|M_C(x_1^*) - M_C(x_2^*)\|_2^2 & similar \quad (25) \\ max(0, margin - \|M_C(x_1^*) - M_C(x_2^*)\|_2)^2 & dissimilar \quad (26) \end{cases}$$

---

[4]The implementation references the official implementation of [Vepakomma et al., 2020]: `https://github.com/tremblerz/nopeek`

$$\min_{M_C, M_S} [L_{util} + \lambda_3 \, L_{Siamese}] \tag{27}$$

, where $x_1^*$ and $x_2^*$ are data points, and the hyperparameter $margin$ is set to control the variance of the feature space, the $\lambda_3$ is set to control the trade-off between utility and privacy.

**Hyperparameters of Defenses** Based on the attack effect of IN (for IN is more stable), we select three sets of hyperparameters for each defense mechanism to achieve varying degrees of privacy protection. In general, under the same defense, the higher the accuracy loss of the model, the better the privacy protection effect, indicating a trade-off between privacy and utility. The accuracy of each defended model and its corresponding defense hyperparameters are shown in Table 3.

Table 3: Details of defense hyperparameters (we set the split point uniformly to Block3). For Shredder, we sample the initial $T_{noise}$ that follows a Laplace distribution parameterized by location 0 and scale 20. Note that Shredder is the only defense mechanism that ***does not require retraining the target model***. We train 50 distributions for Shredder, maintaining an accuracy of over 77% for all of them.

| Defense mechanisms | Hyperparameters | Settings |
|---|---|---|
| Dropout Defense | $r$ | 0.7, 0.8, 0.9 |
| DISCO | $(\rho, R)$ | (0.75,0.2), (0.95,0.1), (0.95,0.5) |
| Noise Mask | $(loc \, a, scale \, b)$ | (0,0.5), (0,1.0), (0,1.5) |
| Shredder | $\alpha = 0.001$ | Randomly select 3 out of 50 distributions. |
| Adversarial Learning | $\lambda_1$ | 1, 2, 3 |
| NoPeek | $\lambda_2$ | 3, 5, 10 |
| Siamese Defense | $\lambda_3, margin = 30$ | 0.003, 0.005, 0.009 |

## C   Results of White-box DRAs

Table 4 shows the performance of our DRAs against the original model in the White-box setting.

## D   Defense Analysis

### D.1   Analysis of NoPeek

We evaluate the performance of *GLASS* against the same image on both the original model and the model after NoPeek. Figure 9 (a) shows that the optimization curve of NoPeek tends to 0 when the number of iterations is small, which means that Feature Obfuscating type defenses represented by NoPeek mainly map different input images to similar intermediate feature representations to realize defense. Figure 9 (b) further confirms the conclusion. It can be seen that the feature loss of two similar reconstructed images is two orders of magnitude different.

### D.2   Comparison of Defenses

As Figure 10 shows, the upper left curve implies a better privacy-utility trade-off. NoPeek and DISCO achieve the optimal defensive effect on almost all DRAs. Most of the curves show an increasing trend, that is, the loss of model accuracy is positively correlated with the privacy defense effect.

Table 4: Reconstruction attacks performance in the White-Box setting. **Red** represents the optimal performance and **Blue** represents the second-best performance.

| Split Point | Method | Evaluation Metrics | | | | |
|---|---|---|---|---|---|---|
| | | LPIPS ↓ | SSIM ↑ | PSNR ↑ | MSE ↓ | NIQE ↓ |
| Block1 | rMLE | 0.042 | 0.961 | 32.88 | 0.003 | 14.50 |
| | LM | 0.046 | 0.956 | 32.64 | 0.003 | 14.38 |
| | IN | **0.026** | 0.961 | 31.98 | 0.003 | **12.76** |
| | GLASS | **0.032** | **0.971** | **36.31** | **0.001** | 14.24 |
| | GLASS++ | 0.037 | **0.967** | **35.60** | **0.001** | **14.18** |
| Block2 | rMLE | 0.258 | 0.733 | 24.26 | 0.022 | 15.73 |
| | LM | 0.131 | 0.874 | 28.24 | 0.009 | 15.36 |
| | IN | 0.248 | 0.757 | 25.13 | 0.017 | 15.85 |
| | GLASS | **0.097** | **0.876** | **28.66** | **0.008** | **13.54** |
| | GLASS++ | **0.065** | **0.924** | **31.56** | **0.004** | **14.00** |
| Block3 | rMLE | 0.415 | 0.608 | 21.43 | 0.041 | 16.54 |
| | LM | 0.298 | 0.723 | 23.71 | 0.024 | 16.42 |
| | IN | 0.289 | 0.685 | 22.86 | 0.029 | 15.73 |
| | GLASS | **0.140** | **0.808** | **25.71** | **0.016** | **13.86** |
| | GLASS++ | **0.128** | **0.833** | **26.56** | **0.014** | **15.30** |
| Block4 | rMLE | 0.638 | 0.326 | 14.31 | 0.233 | 16.56 |
| | LM | 0.536 | 0.517 | 18.81 | 0.077 | 16.91 |
| | IN | 0.295 | 0.632 | 20.66 | 0.050 | 15.75 |
| | GLASS | **0.252** | **0.644** | **21.19** | **0.049** | **14.69** |
| | GLASS++ | **0.183** | **0.736** | **23.15** | **0.031** | **14.26** |
| Block5 | rMLE | 0.764 | 0.215 | 12.11 | 0.351 | 16.32 |
| | LM | 0.842 | 0.225 | 12.11 | 0.357 | **14.11** |
| | IN | 0.377 | **0.476** | **16.14** | **0.149** | **14.95** |
| | GLASS | **0.374** | 0.338 | 12.97 | 0.313 | 15.08 |
| | GLASS++ | **0.293** | **0.499** | **16.13** | **0.157** | 15.27 |
| Block6 | IN | 0.437 | 0.365 | 13.32 | 0.295 | 16.60 |
| | GLASS++ | **0.369** | **0.373** | **14.38** | **0.215** | **15.19** |

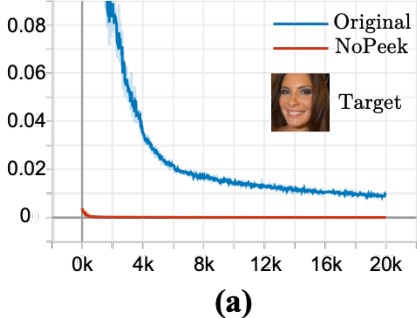

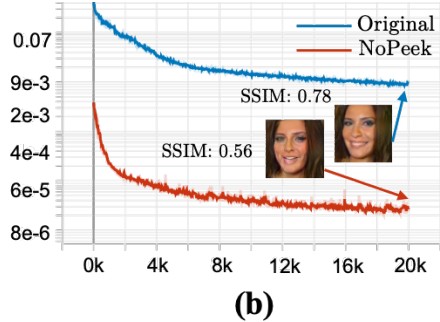

Figure 9: The curve depicting the change of feature loss as the number of optimization iterations increases.

### D.3 Analysis of our DRAs under Defenses

Compared with other DRAs, it can be seen in Figure 11 (a) that the curves corresponding to our methods are more concentrated on the right side (SSIM>0.5), which means that the attacks have higher robustness to the defenses. Moreover, under the attack of our DRAs, the curve corresponding to most defenses becomes relatively vertical, suggesting that we somewhat break the privacy-utility trade-off.

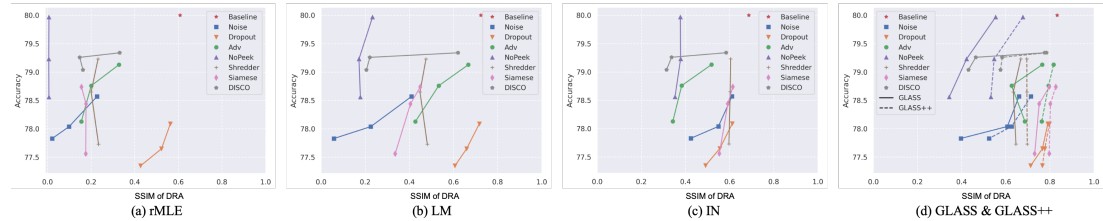

(a) rMLE        (b) LM        (c) IN        (d) GLASS & GLASS++

Figure 10: The correlation between the model's accuracy and the SSIM of DRA under different attacks and defenses.

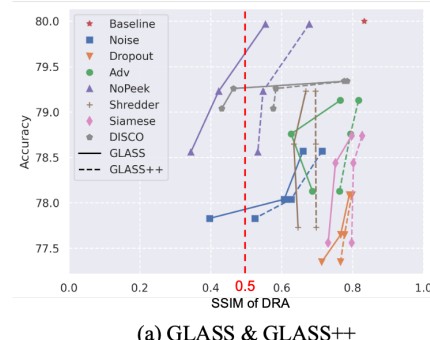

(a) GLASS & GLASS++

Figure 11: The correlation between the model's accuracy and the SSIM of our DRAs under different defenses.

# E  Extended Experiments

## E.1  Black-box & Query-free Settings.

**Black-box Setting**    We relax the assumption of adversary capability and conduct experiments in Black-box and Query-free settings. In the Black-box setting, we apply two gradient-free optimization methods, CMA[Hansen, 2016] and RandomSearch[5], and substitute $\mathcal{W}+$ space search with $\mathcal{W}$ space search to implement optimization. The reason is that gradient-free optimization is relatively weak in high-dimensional space, so our subsequent experiments based on gradient-free optimization are all conducted in $\mathcal{W}$ space. Figure 12 and Table 5 demonstrate that gradient-free optimization can effectively expose sensitive information in a single face, requiring just **2,000** iterations.

Table 5: Reconstruction attacks performance in the Black-Box setting.

| Split Point | Method | Evaluation Metrics | | | | |
| --- | --- | --- | --- | --- | --- | --- |
| | | LPIPS ↓ | SSIM ↑ | PSNR ↑ | MSE ↓ | NIQE ↓ |
| Block2 | CMA | 0.314 | 0.504 | 18.02 | 0.088 | 12.33 |
| | RandomSearch | 0.359 | 0.391 | 15.16 | 0.175 | 12.97 |
| Block3 | CMA | 0.329 | 0.462 | 16.51 | 0.132 | 12.69 |
| | RandomSearch | 0.352 | 0.383 | 14.49 | 0.211 | 13.70 |
| Block4 | CMA | 0.424 | 0.324 | 13.40 | 0.286 | 11.98 |
| | RandomSearch | 0.373 | 0.335 | 12.95 | 0.305 | 13.12 |

---

[5]We apply APIs from https://facebookresearch.github.io/nevergrad/

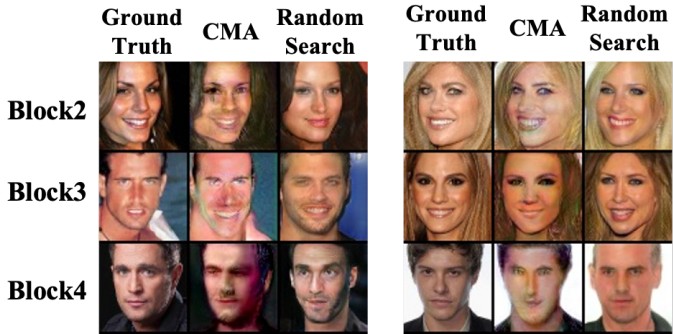

Figure 12: Black-Box *GLASS* under different split points.

**Query-free Setting**     In the Query-free setting, we have made interesting attempts and improvements based on the *shadow model* reconstruction technique proposed by [He et al., 2019]. Firstly, we replace the parameters of the shadow client model $M_C^S$ with the pre-trained ones instead of randomly initialized ones. This pre-trained model is trained by the attacker on the target task with public data. has greatly enhanced the effectiveness of the attack. As shown in Table 6, *GLASS* achieves an SSIM of **0.392**, surpassing the **0.152** of IN. Intuitively, we believe that this improvement is due to the fact that when the pre-trained $M_C^S$ performs fine-tuning on the target task in conjunction with the server model $M_S$, compared with the randomly initialized convolution kernel, the trained convolution kernel can extract more features. This allows it to adapt to $M_S$ for utility while extracting additional features that the regular conventional kernel can capture, which greatly improves the quality of the reconstructed image in color and texture, as shown in Figure 13. We also equip *GLASS++* with a pre-trained *feature2image* module at Block6 to improve the effectiveness of the attack. Furthermore, we attempted to numerically crop the latent code from the $\mathcal{P}+$ space in order to mitigate the distortion of the reconstructed image. This approach guarantees that feature searching can be conducted effectively within a certain range and provides a trade-off between image similarity and feature-matching accuracy. As indicated in Table 6, $\mathcal{P}+$ space cropping results in a decrease in LPIPS from **0.479** to **0.429**. We apply the same measure to the GLASS at Block5.

Table 6: Reconstruction attacks performance in the Query-free setting.

| Split Point | Method | Evaluation Metrics | | | | |
| --- | --- | --- | --- | --- | --- | --- |
| | | LPIPS ↓ | SSIM ↑ | PSNR ↑ | MSE ↓ | NIQE ↓ |
| Block2 | rMLE | 0.747 | 0.085 | 11.66 | 0.381 | 17.25 |
| | LM | 0.738 | 0.096 | 11.21 | 0.426 | 15.52 |
| | IN | 0.718 | 0.152 | 11.78 | 0.371 | **14.17** |
| | **GLASS Fine-tuning** | **0.479** | **0.392** | **14.27** | **0.209** | 14.31 |
| | **GLASS P+ cropping** | **0.429** | **0.329** | **14.04** | **0.230** | **13.25** |

## E.2   Heterogeneous Data.

We suggest that the homogeneity of data has a significant impact on DRA. Specifically, when dealing with homogeneous data, instances exhibit a high level of structural similarity, whereas in the case of heterogeneous data, the level of structural

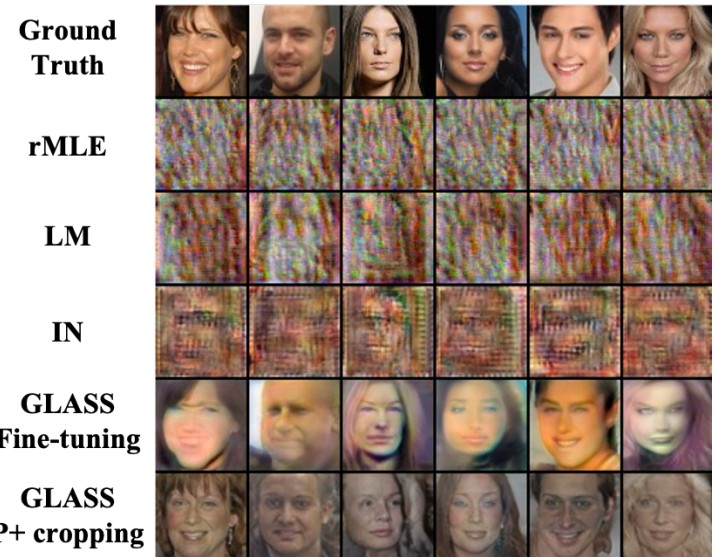

Figure 13: Query-free DRAs under Block2

similarity is comparatively lower. For heterogeneous data, we use (1) CIFAR-10[6] containing 60,000 color images, (2) CINIC-10[7] containing 270,000 color images. Both are $32 \times 32$ images in 10 classes. For CINIC-10, we exclude any images that are identical to those in CIFAR-10 and adopt a pre-trained StyleGAN-XL model[8]. We study DRA against models built for *Image Classification*: 10-class image classification on the CINIC-10. We adopt ResNet-18[He et al., 2016] and split the target model $M_C$ into different layers, as shown in Figure 8.

We employ a gradient-free CMA optimizer to facilitate *GLASS* optimization. This choice is motivated by the fact that gradient-free optimization allows for a more substantial perturbation of the representation space, which is very effective in the reconstruction of heterogeneous data. Figure 14 illustrates that Optimization-based *GLASS* yields superior reconfiguration attack outcomes across multiple Blocks compared to the IN. The reason behind this discrepancy lies in the insufficient ability of the Learning-based method to map features to images with significant structural differences. Consequently, the reconstructed images of the Learning-based DRA tend to become fuzzy. Our method makes up for this shortcoming by leveraging abundant prior knowledge.

## F   t-SNE of Shredder

We further analyze Shredder and find that the intermediate feature representations covered by samples generated from Shredder's noise library can be clearly clustered, even when the noise library contains 50 sets (far exceeds the default of 20) of well-trained distributions, as shown in Figure 15. The specific experimental procedure is as follows. First, we train a Shredder noise library consisting of 50 sets of distributions (collect the fitted distributions and orders of noise elements).

---

[6] https://www.cs.toronto.edu/~kriz/cifar.html
[7] https://github.com/BayesWatch/cinic-10
[8] https://github.com/autonomousvision/stylegan-xl

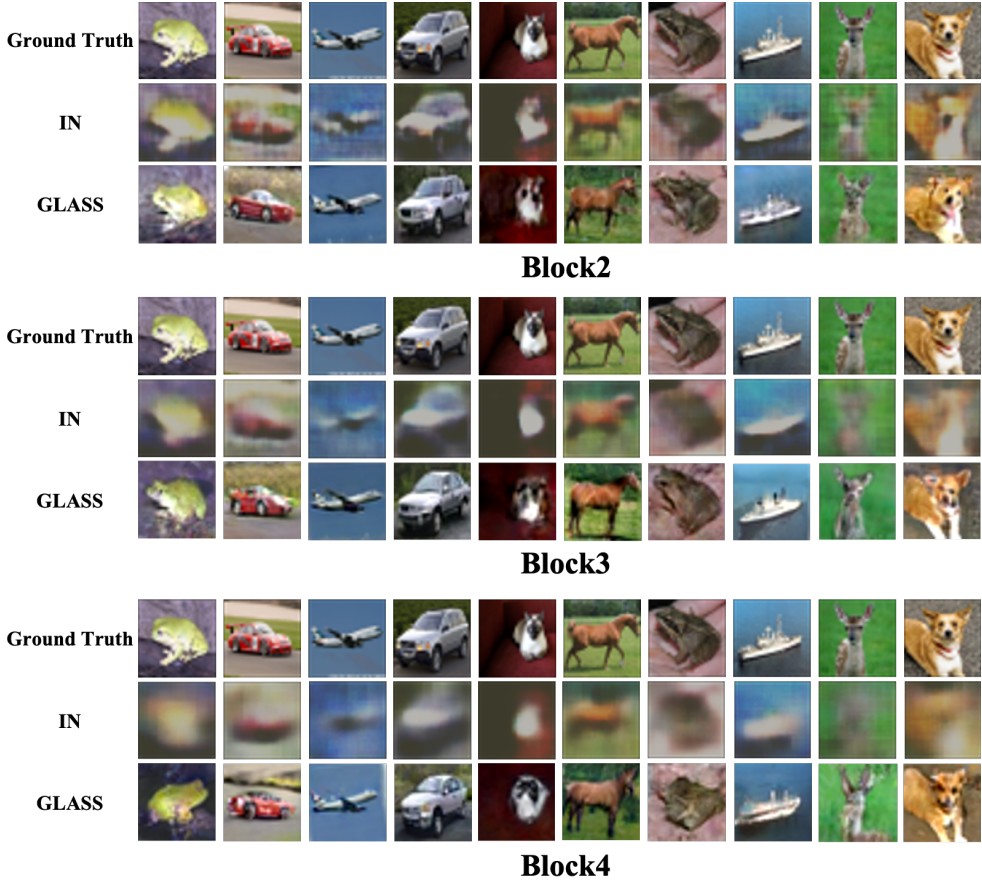

Figure 14: DRAs against Heterogeneous Data.

Then, we randomly sample noise masks and add them to the target intermediate feature representations (derived from privacy inference instances). Finally, we use k-means[9] clustering on these processed features to obtain labels, and then employ t-SNE [Van der Maaten and Hinton, 2008] to reduce their dimensionality and visualize them.

Therefore, for Optimization-based and Learning-based DRAs, we only need to collect intermediate feature representations and utilize those that can be mapped to the same set to carry out DRA. This reduces the Shredder's noise library from multiple distributions to a single distribution.

---

[9]We use the k-means implementation from scikit-learn: https://scikit-learn.org/stable/about.html.

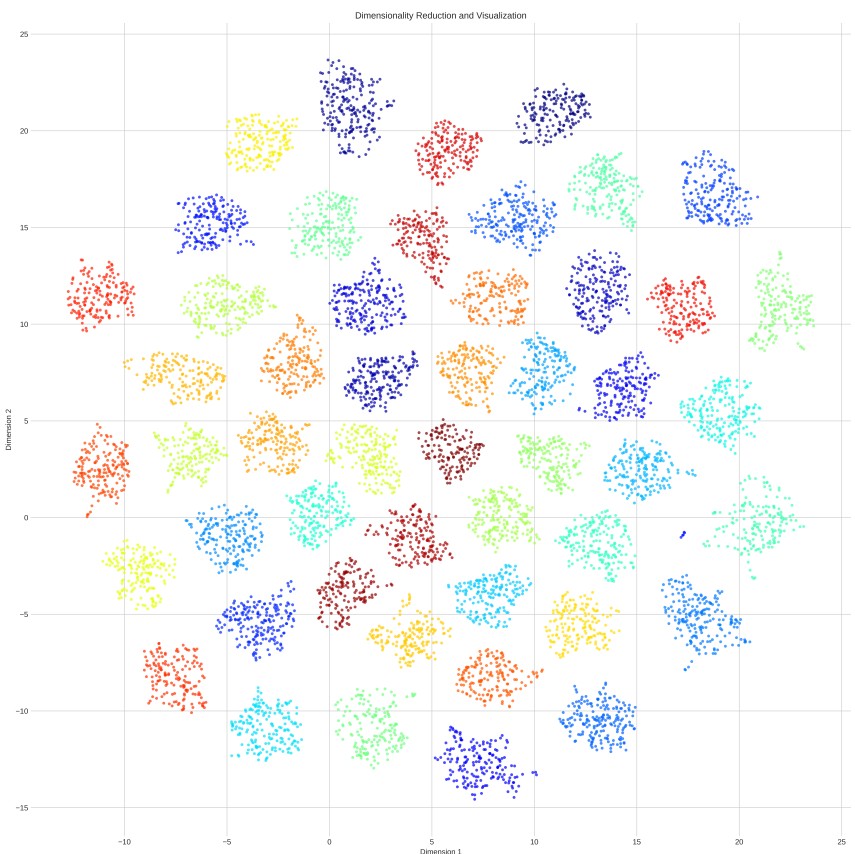

Figure 15: Dimensionality Reduction and Visualization of intermediate feature representations after Shredder processing.