# OpenReview forum: "GAN You See Me? Enhanced Data Reconstruction Attacks against Split Inference"
_NeurIPS.cc/2023/Conference — NeurIPS 2023 poster_

### Official Review · Reviewer_WYrE · 2023-06-26

**Soundness:** 3 good
**Presentation:** 2 fair
**Contribution:** 3 good
**Rating:** 6
**Confidence:** 4

**Summary:**

The paper introduces a new data reconstruction attack (DRA) on split inference (SI) called GLASS and GLASS++. The task of DRAs in SI is to reproduce the image of a certain user based on the intermediate output of the first part of a trained network, which is located on the user's device. GLASS uses StyleGAN as prior knowledge to constrain the generation of images based on given intermediate activations. The paper shows that GLASS is more robust to multiple defense methods against DRAs and that the generated images are of higher quality than previous DRA approaches.

**Strengths:**

- the paper presents a novel DRA attack
- the paper presents a good idea to use StyleGAN as prior knowledge for DRAs
- the paper demonstrates that existing defenses are not effective when using DRAs based on generative models.

**Weaknesses:**

- It is quite hard to follow the paper. For example, it is not clarified what “perturbation” is during the explanation of the approach (paragraph about “Z space search”).
- Setting the learning rate and the number of steps used for the attack to be the same for GLASS, rMLE and LM is not fair in my opinion. Different attacks might need a different number of steps or a different learning rate based on the optimization objective. The comparison would be much fairer when the optimal hyperparameters are taken for each of these attacks.
- There are too many references and experimental results in the supplementary material. It is hard to read the paper without reading the supplementary material. For example, in the abstract, the paper claims that the approach is evaluated on 7 defense methods. However, not all results for the evaluations are shown in the main paper and are instead only present in the appendix.

Misc:
Line 249: Table 4 -> should probably be Table 1
Line 264: Table 5 -> should be Figure 5
Line 203: sigma() is not properly defined

**Questions:**

Q1: What does “searching in Z space for the reason that the entanglement of Z space increases the amplitude of positive perturbation in representation space” mean? Could you elaborate what the “amplitude of positive perturbation” is?

Q2: Related to the previous question, can you elaborate on what exactly you mean by disturbance in line 142?

Q3: Why is the total variation loss required? Shouldn't StyleGAN create realistic images based on the latent variable and as a result, the total variation loss shouldn't be needed?

Q4: In Figure 3, the paper claims that GLASS++ is achieving to find the global optimum. As far as I am aware, it is not possible to prove that the found solution is, in fact, the global optimum. Can you elaborate on how you come to the conclusion that the result is the global optimum instead of a local one?

Q5: It would be interesting to see not only the mean of the metrics, but also the standard deviation. This would give an impression on how consistent the attack results are. Could you state the standard deviation for the experiments in table 1?

**Limitations:**

For the experiments on CelebA the paper reserves roughly 40% of the dataset for performing the evaluation, while the StyleGAN was trained on 60% of the dataset. The assumption that the attacker has 60% of the data from the exact same distribution as the input he is trying to recreate is quite restrictive. The effect of a distribution shift between the image to be reconstructed and the GAN is addressed in the supplementary material. However, the distribution shift is only tested on an undefended model, which is why it is not possible to make a claim about the influence of the shift regarding defended models. Showing results on the defended models with a distribution shift would strengthen the findings of the paper.

---

> ### Author Rebuttal · Authors · 2023-08-09
>
> We highly appreciate the invaluable and perceptive feedback offered by the reviewer. We have considered all the concerns mentioned and responded appropriately to each one.
>
> **Answer for Weakness1, Qustion1, Qustion2, Qustion4:**
>
> Before proceeding, we kindly ask you to consult the detailed information available in the global response, specifically response.pdf/Figure-4. The answers provided below are grounded in this context.
>
> Regarding **Weakness1**, the "perturbation" referenced in the paper pertains to modifications within the representation space, involving alterations in corresponding facial features resulting from latent code optimization. The term "positive perturbation" denotes changes that diminish reconstruction loss and are favorable for an attack.
>
> Regarding **Question1**, the observation of response.pdf/Figure-4(a) reveals that even though the Z space search's trajectory variation amplitude is small, it induces significant facial feature changes, ultimately converging to the target features. This mirrors the greater trajectory variation amplitude in the W space search, with the resultant mapped latent code converging to the target, as depicted in response.pdf/Figure-4(b). This signifies that Z space search enhances the magnitude of facial feature changes and broadens the scope of feature exploration. Furthermore, this adjustment brings the reconstructed outcomes closer to the target image, signifying an "increase in the amplitude of positive perturbation in the representation space."
>
> Regarding **Question2**, we first elaborate on the merits of Z and W+ space searches. W+ space search introduces more nuanced feature alterations, enhancing reconstruction precision. However, its restricted range of feature changes increases the risk of getting stuck in local optima during optimization. The Z space search remedies this limitation by offering larger feature perturbations (controlled disturbances) that serve as advantageous starting points, efficiently sidestepping the W+ space search's potential proximity to local optima.
>
> Regarding **Question4**, for the same reasons detailed above, we favor W space over W+ space for more insightful analysis. As portrayed in response.pdf/Figure-4(c), distinct initializations w1, w2, w3, and w4 lead their respective W space searches to converge on the target latent code. Enhanced initialization and W/W+ space searches in GLASS++ render this process even smoother. This elucidates our conclusion of "the global optimum." Nevertheless, we recognize the need to revise this wording due to inaccuracy. When the intermediate feature information is exceedingly scarce (such as when the split point is set at Block 5/6), achieving the target latent code becomes unfeasible. We will alter "the global optimum" to "an optimum attainable by the attacker with existing knowledge." We apologize for the imprecise phrasing, which will be rectified in our revised version.
>
> **Answer for Weakness2:**
>
> Thank you for your feedback. In our view, optimization-based data reconstruction attacks tend to yield improved attack effectiveness as the number of optimization iterations increases. However, as the optimization process advances, the gains in reconstruction enhancement tend to diminish. To address this, we've set a sufficiently large number of iterations (20,000), ensuring that various attacks converge stably to their attainable optima. As depicted in response.pdf/Figure-5, the feature loss for the three optimization-based attacks exhibits complete convergence during the reconstruction process. To validate this, we adjusted the learning rate from a uniform 1e-2 to 1e-1. As evident in the reconstruction outcomes, the three results underwent minimal alteration, further affirming the robust convergence of the attack outcomes.
>
> **Answer for Weakness3:**
>
> We appreciate your valuable suggestions. We will thoroughly revise our paper to ensure that crucial results and analyses are appropriately incorporated into the main body.
>
> **Answer for Weakness4:**
>
> We apologize for the errors that have occurred. We will conduct a comprehensive review of the entire paper and rectify these issues in the revised version.
>
> **Answer for Qustion3:**
>
> The impact of total variation loss on the attack's effectiveness is generally limited, owing to StyleGAN's proficiency in generating realistic images. Nonetheless, W+ space offers extensive image editing capabilities, potentially causing the latent code to deviate from the sampling distribution of W+ space during optimization. Illustrated in response.pdf/Figure-6, under the Noise Mask defense, GLASS's unrestrained reconstruction exhibits unnatural artifacts. The inclusion of the total variation loss term rectifies this by producing smooth and natural reconstructed images.
>
> **Answer for Qustion5:**
>
> We show the standard deviations of GLASS and GLASS++ in response.pdf/Table-1.
>
> **Answer for Limitation:**
>
> In real-world split inference systems, the common practice involves the server training the model with its own data and then sharing it with clients for collaborative inference through APIs. Our setup aligns well with this scenario, as the server grants API access based on specific criteria for private inference data. To explore a narrower attack scope, a simple solution could be fine-tuning other pre-trained StyleGAN models with a small CelebA image subset. We'll conduct and include these experiments in the revised version.
>
> We present results pertaining to defended models encountering distribution shifts in response.pdf/Figure-7. Although our reconstruction results experienced a moderate decline, the privacy feature disclosure capability remains superior to that of Inverse-Network. While time constraints constrained us to explore only GLASS, we anticipate that the more stable GLASS++ will yield enhanced reconstruction outcomes, which we'll also incorporate in the revised version.

---

> > ### Comment · Reviewer_WYrE · 2023-08-13
> > **Addressing Rebuttal #1**
> >
> > Thank you for the detailed rebuttal!
> > My concerns have been appropriately addressed, which is why I am raising my score from 3 --> 6 to `weak accept`.

---

> > > ### Author Response · Authors · 2023-08-19
> > >
> > > We genuinely appreciate your prompt response to our rebuttal.

---

### Official Review · Reviewer_k9Qp · 2023-07-05

**Soundness:** 2 fair
**Presentation:** 3 good
**Contribution:** 2 fair
**Rating:** 5
**Confidence:** 3

**Summary:**

This paper proposes GAN-based latent space search attack (GLASS) that leverages a pre-trained StyleGAN for reconstructing private data from shared representations in split inference via a two-step search in the Z space and the W+ space. Additionally, GLASS++ is proposed to learn a mapping model to produce better initial points for subsequent optimizations. The effectiveness of the proposed method is evaluated on the CelebA and FFHQ datasets against several defenses.

**Strengths:**

1. The paper is well-structured and easy to follow.

2. It is interesting to consider combining optimization-based attacks with learning-based attacks.

3. The evaluation considered several attack baselines and defense mechanisms.


**Weaknesses:**

1. Some closely related prior works [R1, R2] were not discussed/compared. For instance, the idea of utilizing GAN inversion to improve data reconstruction has been introduced in [R1], which utilizes a pre-trained GAN to invert latent vectors produced by DNN to the corresponding input images through optimization. The usage of a StyleGAN with the latent space search and learned mapping is new but it is unclear if there are other technical novelties.

2. In the evaluation, it is not quite clear which part of the data is used to train the encoder network, which makes it a bit hard to verify the benefits of GLASS++ as claimed in Figure 3. If there’s a distribution shift in the data used for training the encoder, could it still produce an initial point that is better than random initialization?

3. The experiments only considered CelebA as the private dataset and 40 images for evaluating the attacks. As the optimization process is stochastic and sensitive to initialization, it would be better to consider more data samples to validate the effectiveness of the proposed attacks.

[R1] Zhang, Yuheng, et al. "The secret revealer: Generative model-inversion attacks against deep neural networks." CVPR 2020.
[R2] Dong, Xin, et al. "Privacy Vulnerability of Split Computing to Data-Free Model Inversion Attacks." BMVC 2022.


**Questions:**

Which portion of data is used to train the encoder network used in GLASS++? Would it still perform well if both the StyleGAN and the encoder are trained on a different data distribution?

**Limitations:**

The negative societal impact may be stated more explicitly, e.g., via adding a discussion section.

---

> ### Author Rebuttal · Authors · 2023-08-09
>
> We highly appreciate the invaluable and perceptive feedback offered by the reviewer. We have considered all the concerns mentioned and responded appropriately to each one.
>
> **Answer for Weaknesses:**
> 1. We agree that GAN inversion techniques are used in some of the study cases of Model Inversion Attacks (MIAs) and Gradient Inversion Attacks (GIAs) as well. However, our contribution differs significantly from these studies. The innovativeness of our work does not lie in the use of GAN inversion, but rather in exploiting the latent space search characteristic and advantageous disentangled representation of StyleGAN to develop novel Data Reconstruction Attacks (DRAs) specifically targeting Split Inference systems. The MIAs represented by [R1] aim at obtaining sensitive image features of individuals in the original training data based on the coupled feature information contained in the confidence scores of the face ID classification model, while DRAs focus on reconstructing the client private inference data by utilizing intermediate feature representations outputted from the splitting layer of DNN models with different functionalities. Regarding [R2], this work is methodologically similar to the baseline Inverse Network (IN) that we have compared. We will provide additional comparative experiments or include discussions of these closely related prior works in our manuscript.
>
> 2. We thank the reviewer for the comment. We agree that our description of the training data for the encoder network was not very clear. In all GLASS++ experiments, the training data of the encoder network is the same as the data used to train the StyleGAN model used by the attacker. Compared with GLASS, GLASS++ only increases the cost of the attacker's computing resources. In the experiments presented in Appendix D.3, we substituted the StyleGAN model used in the attack with one trained on the FFHQ dataset, instead of CelebA, and utilized FFHQ data to train the encoder network. The results demonstrate that even with a shift of data distribution, GLASS++ still has an effective reconstruction attack effect. That is, even if there’s a distribution shift in the data used for training the encoder, it can still produce a better initial point. We will provide detailed description and extended ablation study in our revision.
>
> 3. Thanks for the comment. We would like to clarify that we have extended our method to heterogeneous data (CINIC-10 as the private dataset), which is mentioned in Lines 308-314. The experiments in Appendix D.2 demonstrate the adaptability of our method. We believe that the number of samples can validate the effectiveness of our method, but we agree that more samples would help improve the overall quality of our work. We will provide more attack results as well as standard deviation data for relevant experiments in the revised version to demonstrate the stability of our attack effects.
>
> **Answer for Questions:**
>
> Please refer to our answer for **Weaknesses-2**.
>
> **Answer for Limitations:**
>
> We thank the reviewer for this suggestion. We will provide additional discussions with respect to the ethical implications of enabling more effective DRAs in our manuscript. The potential negative societal impact of GAN-based DRAs mainly stems from the disclosure of private data. Once the attacker reconstructs the original data fed into the DL model at the edge-side, it can lead to privacy invasion and generate malicious false information. We hope that our proposed attacks will draw attention to the privacy protection of split inference systems and promote the development of more effective defense mechanisms.
>
> *References:*
>
> [R1] Zhang, Yuheng, et al. "The secret revealer: Generative model-inversion attacks against deep neural networks." CVPR 2020.
>
> [R2] Dong, Xin, et al. "Privacy Vulnerability of Split Computing to Data-Free Model Inversion Attacks." BMVC 2022.

---

> > ### Comment · Reviewer_k9Qp · 2023-08-21
> > **Thank you**
> >
> > Dear authors,
> >
> > Thank you for your prompt response, which the reviewer greatly appreciates. The suggestion is to consider integrating the relevant discussion and experiment details into the revised version to improve clarity.

---

> > > ### Author Response · Authors · 2023-08-21
> > >
> > > Your feedback on our rebuttal is highly valued. We will enhance our paper by incorporating your valuable suggestions.

---

### Official Review · Reviewer_BEtd · 2023-07-07

**Soundness:** 3 good
**Presentation:** 3 good
**Contribution:** 3 good
**Rating:** 6
**Confidence:** 3

**Summary:**

The paper proposed GLASS and GLASS++, which utilize StyleGAN to launch data reconstruction attacks against Split Inference. This is the first GAN-based reconstruction attacks against split inference, and it shows consistently better results compared with previous methods, against 7 defense schemes.

**Strengths:**

1. As stated by authors, this is the first GAN-based reconstruction attacks against SI. I am not sure if similar ideas (search GAN latent space in privacy attacks) have been proposed in similar topics (for example other privacy attacks like gradient inversion or model inversion), so I would like to discuss with other reviewers about the novelty or originality of the method.

2. The work has achieved a high level of completion, and the experiments were conducted comprehensively. The results are reported to be state-of-the-art for nearly every setting.

**Weaknesses:**

The results highly rely on the (1) similarity of auxiliary distribution and private distribution and (2) The complexity of the distribution modeled.  Although the authors discussed the performance under distribution shift, they conducted experiments on FFHQ and CelebA, which are both face image datasets, well aligned and structured.  In practice, the server may not know much about data distribution from end users, so the auxiliary dataset and private dataset could have significantly different distributions. Additionally, in practice, the distribution could be highly complex, for example, for a facial recognition system, photos from end users are not likely to be cropped and well-aligned. They may have various backgrounds, poses, and lighting conditions. Although the paper provides SOTA methods over their settings, it is worth discussing how the method will perform under more challenging settings.

**Questions:**

1. Could you provide some visualizations on the intermediate results of searching the optimal W+, to make readers understand more about the optimization progress and how your model helps the optimization towards optimal reconstruction?


**Limitations:**

Same as weakness.

---

> ### Author Rebuttal · Authors · 2023-08-09
>
> We deeply value the priceless and insightful feedback provided by the reviewer. We have taken into account all of the mentioned concerns and addressed them accordingly.
>
> **Answer for Weaknesses:**
>
> We thank the reviewer for the comment and would like to further clarify the adversary's knowledge about the data distribution. In real-world protocols for Split Inference, it is relatively easy for a server-side adversary to know the tasks of the target model and the information related to the data distribution, since the service provider usually obtains the complete target model and deploys it in a split. This knowledge allows the adversary to choose auxiliary data with a distribution similar to the client-side private inference data. In addition, existing privacy attacks and defenses studies in machine learning [1][2][3][4][5] typically use aligned/structured face images as assumptions for private data and auxiliary data, and our work follows this setup as well.
>
> Our method is also expanded to heterogeneous data, as mentioned in Lines 308-314. In Appendix D.2, we illustrate that our approach can be adapted to heterogeneous image data with a lower level of structural similarity. Our experiments involve the utilization of a released StyleGAN-XL model [7], pre-trained on CIFAR-10 [8], to reconstruct private inference data from CINIC-10 [9]. Despite there is a shift in data distribution between CIFAR-10 and CINIC-10, our attack demonstrates effectiveness in this setting, as depicted in Figure 12. In Lines 652-656, an analysis is presented explaining the better performance of our method compared to other DRA. We are sorry for placing this important result in the appendix and assure you that we will make adjustments to the paper's content and structure to include relevant results in the main body.
>
> We concur that a more realistic implementation of DRA on uncropped/unaligned private inference data will enhance the quality of our research. According to [6], we improve StyleGAN by modifying its first-layer feature from constant to variable. Furthermore, we combine it with the latent code of W+ space and carry out joint optimization during the second stage of our method. The evaluation of GLASS shows our method's effectiveness even when different transformations are applied to private inference data, as illustrated in response.pdf/Figure-1. Furthermore, we implement GLASS in a more realistic setting. As shown in response.pdf/Figure-2, we photographed a volunteer and obtained multiple nature images as private inference data. The attack results demonstrate the robustness of our method. It is essential to clarify that in the above experiments, the StyleGAN generator used in our attack was still trained on CelebA which is cropped/aligned. We believe incorporating data augmentation in StyleGAN training could further improve the attack's effectiveness. A dedicated section will be included in our revised version to comprehensively discuss and present the experiments.
>
> **Answer for Questions:**
>
> We thank the reviewer for this suggestion. As shown in response.pdf/Figure-3, based on a good initialization (brought by the Z space search or the encoder network), the W+ space search carries out a fine-grained search for sensitive features. Visually, as the number of iterations increases, the reconstructed image gradually resembles the target image. We will likewise provide clearer and more intuitive explanations in our manuscripts.
>
> **Answer for Limitations:**
>
> Please refer to our answer for **Weaknesses**.
>
> *References:*
>
> Please refer to global response.

---

### Official Review · Reviewer_HnH3 · 2023-07-27

**Soundness:** 3 good
**Presentation:** 3 good
**Contribution:** 3 good
**Rating:** 5
**Confidence:** 3

**Summary:**

The paper titled "GAN You See Me? Enhanced Data Reconstruction Attacks against Split Inference" investigates and proposes new methods of data reconstruction attacks (DRAs) against split inference (SI), a deep learning paradigm that addresses computational constraints on edge devices while preserving data privacy. The authors present GLASS and GLASS++, which are the first DRA methods that use Generative Adversarial Networks (GANs), specifically leveraging StyleGAN, for the purpose of data reconstruction in SI. These methods are evaluated against seven advanced defense mechanisms in the SI paradigm and are found to be effective even in their presence. The authors claim their proposed methods significantly outperform existing DRAs.

**Strengths:**

Originality: The paper introduces the novel application of GANs in data reconstruction attacks, marking a significant shift in the approach to DRAs in SI.

Quality: The paper is technically sound and contains thorough experimental evaluations. It systematically evaluates the proposed methods across different split points, multiple defense mechanisms, and various adversarial settings, providing a well-rounded view of their performance.

Clarity: The paper is well-structured and coherent, with clear descriptions of the problem, proposed methods, and the results obtained. It does an excellent job of explaining the limitations of existing DRAs and how the proposed methods overcome these.

Significance: The work is of high significance as it highlights the existing vulnerabilities of SI, suggesting that even advanced defense mechanisms may not provide sufficient protection against data privacy attacks. This research could lead to more robust defense strategies in SI systems.

**Weaknesses:**

For the paper weaknesses. I think the biggest concern that I have is that the method relies on StyleGAN which is trained on cropped/aligned face data images. One big assumption made here is that the private inference data resembles these cropped/aligned face images. To me, this is a pretty restrictive setting. In order to generalize to other types of image data, the StyleGAN discussed in the paper would not be sufficient. The whole pipeline needs to be redesigned because the pipeline is specially designed with face-StyleGAN in place and certain modules are designed to extract desired styles w+.

The impracticality to generalize and the relatively restrictive setting of the proposed pipeline is the major weakness in my opinion.

In addition, the performance of GLASS and GLASS++ against defense mechanisms is analyzed, but the paper lacks a clear discussion on possible countermeasures or ways to further improve these defenses in light of the presented attacks.

**Questions:**

Please provide responses for my concern in the weakness section above.

Also, please elaborate more on the potential countermeasures that could be implemented to prevent or mitigate the effects of attacks such as GLASS and GLASS++?

**Limitations:**

While the paper showcases impressive results, it seems to lack a comprehensive discussion about the potential ethical implications of enabling more effective data reconstruction attacks, which could be used maliciously.

---

> ### Author Rebuttal · Authors · 2023-08-09
>
> We highly appreciate the invaluable and perceptive feedback offered by the reviewer. We have considered all the concerns mentioned and responded appropriately to each one.
>
> **Answer for Weaknesses:**
>
> - Data Types
>
> We appreciate the reviewer's comment. We'd like to clarify that our attack pipeline isn't solely tailored for face-StyleGAN; rather, we extend our method to diverse data types, as mentioned in Lines 308-314 of the paper. In Appendix D.2, we showcase how our approach adapts to heterogeneous image data with varying structural similarity. Our experiments employed the StyleGAN-XL model [7], pretrained on CIFAR-10 [8], to target private inference data from CINIC-10 [9]. Despite a shift in data distribution between CIFAR-10 and CINIC-10, our attack remains effective, as illustrated in Figure 12. In Lines 652-656, we delve into the reasons for our method's superiority over other DRA methods. We apologize for placing this vital result in the Appendix and assure you that we'll revise the paper's content and structure to incorporate these pertinent findings into the main body.
>
> We acknowledge the challenge associated with generating out-of-range images using StyleGAN. Nevertheless, our research primarily concentrates on harnessing the unique characteristics of StyleGAN's latent spaces to amplify the attack performance of DRA. Our focus lies in refining the accuracy of searching for sensitive privacy features. In the existing landscape of privacy attacks and defenses in machine learning [1][2][3][4][5], it's a prevailing convention to assume the utilization of cropped/aligned face images as private inference and auxiliary data. Adhering to the adversarial settings established in previous work [5], our goal is to unveil greater private information from inference data when compared to baseline methods. With regards to the limitation posed by StyleGAN's reliance on the cropped/aligned faces it's pretrained on, we believe this concern extends to other domains within computer vision. Recent research [6], conducted by Yang et al., has caught our attention, as it briefly investigates the fixed-crop constraint of StyleGAN2 – the primary generative model employed in our experiments. The proposed approach effectively expands the generative scope beyond cropped/aligned faces.
>
> While addressing the aforementioned limitation is not our primary research focus, we concur that a more practical implementation of DRA on uncropped/unaligned private inference data would enhance the quality of our work. According to [6], we enhance StyleGAN by transitioning its constant first-layer feature to a variable one. Furthermore, we integrate this with the latent code of W+ space and undertake joint optimization during the second stage of our methodology. As demonstrated in response.pdf/Figure-1, the evaluation of GLASS underscores our method's effectiveness even when diverse transformations are applied to private inference data. Moreover, we've implemented GLASS within a more authentic context. Illustrated in response.pdf/Figure-2, we capture images of a volunteer and gather multiple nature images as private inference data. The resulting attack outcomes affirm the robustness of our approach. It's crucial to clarify that, in the aforementioned experiments, the StyleGAN generator utilized for our attack was still trained on CelebA, featuring cropped/aligned images. We believe the integration of data augmentation into StyleGAN training could further bolster the attack's effectiveness. In our revised version, we will incorporate a dedicated section to comprehensively discuss and present these experiments.
>
> - Possible Countermeasures
>
> Current defense mechanisms focus on safeguarding entire images from the reconstruction. In contrast, our proposal suggests a targeted approach, concentrating on specific sensitive attributes to enhance defense efficacy. For instance, by masking the mouth region of the input image, we can thwart the attacker's ability to reconstruct this portion, albeit introducing a new trade-off in utility. Furthermore, incorporating adversarial samples against StyleGAN, like generating optimized noise perturbations for each intermediate feature, could potentially yield a more potent defense strategy due to its one-to-one correspondence. We will delve into these aspects in greater detail in the revised version. We sincerely appreciate the reviewer's invaluable feedback, and we are fully committed to enhancing our work based on constructive suggestions.
>
> **Answer for Questions:**
>
> Please refer to our answer for **Weaknesses**.
>
> **Answer for Limitations:**
>
> We appreciate the reviewer's suggestion and will incorporate further discussions regarding the ethical implications of enhancing the effectiveness of DRAs in our manuscript. The potential adverse societal consequences of GAN-based DRAs largely revolve around the exposure of private data. When the attacker successfully reconstructs the initial data supplied to the edge-side DL model, it can result in privacy breaches and the propagation of harmful false information. We anticipate that our proposed attacks will spotlight the significance of safeguarding privacy in split inference systems and encourage the advancement of more robust defense mechanisms.
>
> *References:*
>
> Kindly refer to the comprehensive global response provided earlier.

---

> > ### Author Response · Authors · 2023-08-19
> >
> > If you have any further concerns or questions, please feel free to contact us. We greatly value your feedback.

---

> > ### Comment · Reviewer_HnH3 · 2023-08-21
> >
> > Thanks for the detailed response. I would like to raise my rating. The manuscript can be further improved by incorporating various suggestions by all the reviewers.

---

> > > ### Author Response · Authors · 2023-08-21
> > >
> > > We genuinely appreciate your feedback on our rebuttal. We will truly improve our paper by integrating suggestions from all the reviewers.

---

### Author Rebuttal · Authors · 2023-08-09

The **response.pdf** contains our supplementary experiments.

Here is our detailed explanation of the experiment in the pdf:

**Detailed explanation of response.pdf/Figure-4**:

The Z space is entangled, signifying that even a small change in the latent code within the Z space can yield a large change within the representation space. In other words, a continuous change in the latent code results in an abrupt change in facial features. Conversely, within the relatively disentangled W/W+ space, continuous changes in the latent code lead to continuous changes in facial features. The characteristics of different latent spaces provide them with distinct advantages.

Taking face reconstruction attack as an example. In the optimization process of the Z space search, gradient descent makes the facial features corresponding to the latent code closer to the target features, that is, makes the image generated by StyleGAN closer to the privacy inference image.

Specifically, we use PCA to reduce the dimensionality of the latent codes within latent spaces and visualize them. We use W space instead of W+ space in this example because the latent code of W space has a lower dimensionality (1\*512 compared to 10\*512 of W+ space), which makes dimensionality reduction easier. As shown in response.pdf/Figure-4(a), we initialize z1 and z2 from a normal distribution and perform Z space search to reconstruct the target image. The grey data points correspond to other samples obtained from the normal distribution after dimensionality reduction, whereas the red star symbolizes the reconstructed target image. It should be noted that the target image in this example is selected to be the image generated by StyleGAN, because it ensures that the target latent code is accurate. It can be seen that after the Z space search, the reconstruction results starting with z1 and z2 closely resemble the target image. However, after dimensionality reduction, they appear in three different regions. This is due to the entanglement of the Z space, the same combination of facial features may correspond to multiple different latent codes. We record the latent codes during the Z space search, map them to W space through StyleGAN's mapping network, reduce dimensionality again and visualize. As shown in response.pdf/Figure-4(b), the optimized trajectories within W space corresponding to z1 and z2 finally converge to the target latent code. This convergence can be attributed to the disentanglement of the W space, ensuring that the same combination of facial features corresponds to the same latent code.

As shown in response.pdf/Figure-4(c), for distinct initializations w1, w2, w3, and w4, their respective W space searches converge together to the target latent code.

*References*:

[1] Pasquini, Dario, Giuseppe Ateniese, and Massimo Bernaschi. "Unleashing the tiger: Inference attacks on split learning." Proceedings of the 2021 ACM SIGSAC Conference on Computer and Communications Security. 2021.

[2] Singh, Abhishek, et al. "Disco: Dynamic and invariant sensitive channel obfuscation for deep neural networks." Proceedings of the IEEE/CVF Conference on Computer Vision and Pattern Recognition. 2021.

[3] Vepakomma, Praneeth, et al. "NoPeek: Information leakage reduction to share activations in distributed deep learning." 2020 International Conference on Data Mining Workshops (ICDMW). IEEE, 2020.

[4] Kahla, Mostafa, et al. "Label-only model inversion attacks via boundary repulsion." Proceedings of the IEEE/CVF Conference on Computer Vision and Pattern Recognition. 2022.

[5] Chen, Si, et al. "Knowledge-enriched distributional model inversion attacks." Proceedings of the IEEE/CVF international conference on computer vision. 2021.

[6] Shuai Yang, Liming Jiang, Ziwei Liu, and Chen Change Loy. Styleganex: Stylegan-based manipulation beyond cropped aligned faces. In ICCV, 2023.

[7] Sauer, Axel, Katja Schwarz, and Andreas Geiger. "Stylegan-xl: Scaling stylegan to large diverse datasets." ACM SIGGRAPH 2022 conference proceedings. 2022.

[8] https://www.cs.toronto.edu/~kriz/cifar.html

[9] Luke Nicholas Darlow, Elliot J. Crowley, Antreas Antoniou, and Amos J. Storkey. CINIC-10 is not ImageNet or CIFAR- 10. CoRR abs/1810.03505, 2018. 5

---

### Decision · Program_Chairs · 2023-09-21

**Decision:**

Accept (poster)

**Comment:**

All reviewers lean toward acceptance of the paper. HnH3 raises questions on restrictiveness of the settings and on countermeasures, which are both answered effectively in the rebuttal leading to an increase in scores. The intermediate results presented in the response PDF for BEtd are illustrative and authors are encouraged to include in the main paper. Please include the additional comparisons for use of GAN inversion in MIA and GIA as suggested by k9Qp. While WYrE initially leaned to reject, the rebuttal comprehensively answers the questions leading to an accept rating. Please include the suggested ablations on learning rates in the final supplementary material, the standard deviations in the main paper and results on distribution shifts in the main paper if space permits. In summary, the meta-reviewer agrees with the review consensus to recommend acceptance.